# Mapping of sequences in the 5' region and 3' UTR of tomato ringspot virus RNA2 that facilitate cap-independent translation of reporter transcripts *in vitro*

**Dinesh Babu Paudel**[1¤]**, Hélène Sanfaçon**[2]*

**1** Department of Botany, University of British Columbia, Vancouver, BC, Canada, **2** Summerland Research and Development Centre, Agriculture and Agri-Food Canada, Summerland, BC, Canada

¤ Current address: Saskatoon Research and Development Center, Agriculture and Agri-Food Canada, Saskatoon, Saskatchewan, Canada

* helene.sanfacon@canada.ca

**Data Availability Statement:** All relevant data are within the manuscript and its Supporting Information files

## Abstract

Tomato ringspot virus (ToRSV, genus *Nepovirus*, family *Secoviridae*, order *Picornavirales*) is a bipartite positive-strand RNA virus, with each RNA encoding one large polyprotein. ToRSV RNAs are linked to a 5'-viral genome-linked protein (VPg) and have a 3' polyA tail, suggesting a non-canonical cap-independent translation initiation mechanism. The 3' untranslated regions (UTRs) of RNA1 and RNA2 are unusually long (~1.5 kb) and share several large stretches of sequence identities. Several putative in-frame start codons are present in the 5' regions of the viral RNAs, which are also highly conserved between the two RNAs. Using reporter transcripts containing the 5' region and 3' UTR of the RNA2 of ToRSV Rasp1 isolate (ToRSV-Rasp1) and *in vitro* wheat germ extract translation assays, we provide evidence that translation initiates exclusively at the first AUG, in spite of a poor codon context. We also show that both the 5' region and 3' UTR of RNA2 are required for efficient cap-independent translation of these transcripts. We identify translation-enhancing elements in the 5' proximal coding region of the RNA2 polyprotein and in the RNA2 3' UTR. Cap-dependent translation of control reporter transcripts was inhibited when RNAs consisting of the RNA2 3' UTR were supplied *in trans*. Taken together, our results suggest the presence of a CITE in the ToRSV-Rasp1 RNA2 3' UTR that recruits one or several translation factors and facilitates efficient cap-independent translation together with the 5' region of the RNA. Non-overlapping deletion mutagenesis delineated the putative CITE to a 200 nts segment (nts 773–972) of the 1547 nt long 3' UTR. We conclude that the general mechanism of ToRSV RNA2 translation initiation is similar to that previously reported for the RNAs of blackcurrant reversion virus, another nepovirus. However, the position, sequence and predicted structures of the translation-enhancing elements differed between the two viruses.

**Funding:** The work was supported by Agriculture and Agri-Food Canada (https://www.agr.gc.ca/eng/agriculture-and-agri-food-canada/?id=1395690825741) A-base core funding awarded to HS. The work was also supported in part by the Natural Sciences and Engineering Research Council of Canada (NSERC, https://www.nserc-crsng.gc.ca/index_eng.asp) Discovery Grant RGPIN 122249-10 awarded to HS (and used for a graduate stipend for DBP). The funders had no role in study design, data collection and analysis, decision to publish, or preparation of the manuscript.

## Introduction

Translation of viral RNAs depends on the host translational machinery [1–3]. Eukaryotic cellular mRNAs have a $m^7$G-cap structure at their 5' end (5' cap) and a polyA tail at their 3' end, both known to enhance translation [4, 5]. Canonical translation initiation of cellular mRNAs requires an initial interaction between the mRNA 5' cap and the cap-binding protein (eIF4E), which is part of the eIF4F complex that also contains a scaffold protein (eIF4G) and a loosely associated RNA helicase (eIF4A) [6, 7]. Interaction of eIF4G with the polyA-binding protein (PABP) allows circularization of the mRNA and facilitates translation initiation [8, 9]. The eIF4F complex mediates the recruitment of additional translation initiation factors and helps assemble the ribosomal subunits at an AUG start codon to initiate translation [4, 5]. In plants, isoforms eIF(iso)4E and eIF(iso)4G co-exist with eIF4E and eIF4G and form the alternative complex eIF(iso)4F [10, 11]. While translation normally initiates at the first AUG, the efficiency of translation initiation is affected by the nucleotide sequence that flanks the start codon (termed start codon context) and by the presence of secondary structures near or at the start codon [12–17]. Consensus start codon context for optimal translation initiation have been described for plant and animal mRNAs [14–19].

Although they serve as mRNAs, the genomic and subgenomic RNAs of many positive-sense single-stranded RNA viruses lack the 5' cap structure and/or the 3' polyA tail. *Cis*-acting elements such an internal ribosome entry sites (IRESs) or cap-independent translation enhancers (CITEs) have been shown to recruit components of the host translation machinery (eIF4F, eIF4E, eIF4G or ribosome subunits) to the viral RNA for efficient translation [1–3, 20]. IRES elements are located upstream of the start codon, either in the 5' UTR or in the intercistronic region of dicistronic viral RNAs [2, 3, 21–23]. Most characterized IRESs were identified in the RNAs of animal and human viruses (notably picornaviruses and related viruses from the order *Picornavirales*), whereas only a few have been described for plant viruses, mostly members of the family *Potyviridae* [24]. In contrast to the large highly structured animal and human virus IRESs, most characterized potyvirid IRES are relatively unstructured, with the notable exception of the triticum mosaic virus IRES [24].

CITEs have been mostly described from the 3' UTRs of plant viral RNAs, in particular from the RNAs of viruses in the family *Tombusviridae* or the genus *Luteovirus* that lack both a 5' cap and a 3' polyA tail [20, 25]. Different types of CITEs have been characterized that differ in their secondary structure and in their ability to recruit eIF4E, eIF4G or ribosome subunits [20, 25–27]. To enhance translation, 3' CITE elements normally require the presence of sequences and/or structural elements in the cognate 5' UTR. Long-distance RNA-RNA interactions between the 3' CITE and the 5' UTR, such as kissing loops, have been identified for many plant viral RNAs and probably enhance translation by allowing RNA circularization and the shuttling of translation factors from the 3' CITE to the 5' end of the RNA [20, 25]. In the case of turnip crinkle virus, it has been suggested that RNA circularization is promoted by a protein bridge formed by the recruitment of different ribosome subunits by the 3' CITE and by a polypyrimidine region in the 5' UTR [28, 29].

Tomato ringspot virus (ToRSV) belongs to the genus *Nepovirus*, family *Secoviridae*, order *Picornavirales* [30–32]. ToRSV has a bipartite genome. Like other members of the order *Picornavirales*, the ToRSV viral RNAs are polyadenylated at their 3' ends and are covalently linked to a small viral protein (VPg) at their 5' ends. The polyproteins translated from RNA1 and RNA2 are processed by the viral protease to produce replication and structural proteins, respectively [33–36]. The N-terminal protein domains from the RNA1 and RNA2 polyproteins are not well characterized and the translation initiation sites for the polyproteins have not been experimentally confirmed. The viral 3' UTRs are long (~1550 nts) but the predicted 5'

UTRs of the genomic segments are shorter (76 nts for the RNA2 of ToRSV Rasp1 isolate, assuming that translation initiates at the first AUG) [37, 38]. The viral 5' and 3' UTRs show extensive sequence identity between the genomic RNAs and also amongst isolates of the virus, with 5' terminal sequence identity extending into the coding region [39, 40].

Translation initiation mechanisms of nepovirus RNAs are generally not well understood [31]. Nepoviruses are genetically very diverse and are divided into three subgroups based on serological relationships and on the size of their RNAs [31, 32]. Subgroup C nepoviruses, such as ToRSV, are characterized by their very long 3' UTRs (1300–1600 nts), while subgroup A and B nepoviruses have shorter 3' UTRs (200–400 nts). Using reporter transcripts that include the viral 5' and 3' UTRs of blackcurrant reversion virus (BRV, a subgroup C nepovirus), Karetnikov and colleagues showed that both UTRs are required for efficient cap-independent translation [41–43]. The region of the 3' UTR immediately downstream of the polyprotein stop codon was found to be essential for translation and was suggested to contain a CITE, although other regions of the 3' UTR were also shown to contribute to translation efficiency, at least for RNA2 [41, 42]. An IRES-like sequence in the BRV 5' UTR directed efficient translation initiation when inserted between open reading frames in bicistronic transcripts, but differed from other characterized IRESs because it required the presence of the viral 3' UTR to function [41, 43]. It was proposed that translation initiation of BRV RNAs depends on a split-IRES with functional elements in both the 5' and 3' UTRs [3]. Based on limited sequence identities found in the 5' and 3' UTRs of BRV and of other nepoviruses (including ToRSV), it was suggested that other nepoviruses adopt translation initiation mechanisms similar to those employed by BRV [41–43]. However, this concept has not been investigated experimentally.

Here, we used reporter transcripts containing the 5' region and 3' UTR of the RNA2 of ToRSV Rasp1 isolate (ToRSV-Rasp1) to confirm that translation initiates at the first AUG. We show that sequences and/or secondary structure elements in the 5' region of the RNA and in the 3' UTR contribute to efficient cap-independent translation of the reporter transcripts. We also provide evidence for the presence of a CITE in the 3' UTR. Deletion mutagenesis delineated a 200 nts segment in the distal region of the 3' UTR that is essential for efficient translation in conjunction with the 5' region of the RNA. In contrast to BRV reporter transcripts, deletion of the region of the 3' UTR immediately downstream of the stop codon did not significantly reduce the translation rate. Thus, our results highlight both similarities and differences between BRV and ToRSV translation mechanisms.

## Materials and methods

### Preparation of reporter constructs

All luciferase reporter constructs incorporated the renilla luciferase (Rluc) open reading frame (ORF) (Genbank accession: KJ140114) (**Fig 1**). The VRV (<u>V</u>iral 5' region–<u>R</u>luc ORF–<u>V</u>iral 3' UTR) construct includes the first 443 nts of the 5' region of ToRSV-Rasp1 RNA2 (Genbank accession: KM083895) placed upstream of the Rluc ORF and the entire 3' UTR (1547 nts long, nts 6017–7563) of ToRSV-Rasp1 RNA2 positioned downstream of the Rluc ORF. Similarly, the control reporter construct ARA (<u>A</u>ctin 5' UTR–<u>R</u>luc ORF–<u>A</u>ctin 3' UTR) consists of the Rluc ORF flanked by the 5' UTR (197 nts) and 3' UTR (366 nts) from the *Arabidopsis thaliana* actin gene (Genbank accession: NM103814). A 12 nts linker containing a BamHI restriction site was inserted immediately downstream of the actin 5' UTR. The cloning of Rluc ORF downstream of the 5' UTR (from actin or viral) created a unique BsrDI restriction site. To facilitate the cloning of subsequent derivatives of these constructs, a short linker sequence containing an EcoRV restriction site was inserted between the Rluc and the 3' UTR (from actin or viral). Also, a second linker sequence with an XbaI restriction site was placed between the 3'

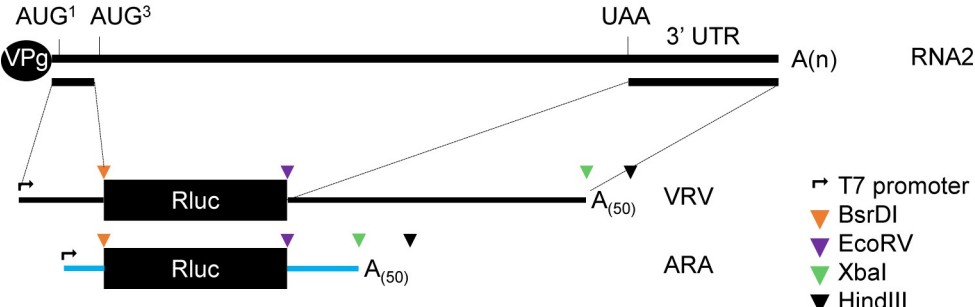

**Fig 1. Schematic representation of reporter constructs.** All constructs include the Rluc (Renilla luciferase) reporter gene and a T7 promoter site upstream of the 5' UTR for *in vitro* transcription. The ToRSV-Rasp1 RNA2 is shown at the top along with the AUG[1] and AUG[3] putative start codons and the UAA stop codon. The VRV reporter transcript consists of the Rluc ORF (black box) flanked by the 5' region (first 443 nts) and complete 3' UTR of ToRSV-Rasp1 RNA2 (black lines). The control ARA transcript, includes the 5' and 3' UTRs from the *Arabidopsis thaliana* actin gene (blue lines). A 50 nts stretch of polyA sequence followed by a HindIII restriction site was added at the 3' end of all constructs. Additional restriction sites that facilitate cloning of derivative constructs are shown by coloured triangles.

UTR and a 50 nts polyA sequence. All reporter constructs were assembled under the control of a T7 promoter to facilitate *in vitro* transcription starting from the first nucleotide of the 5' UTR.

The VRV and ARA constructs were synthesized commercially by GeneArt (Thermo Fisher) and were inserted in the KpnI-HindIII restriction site of plasmid pMK-polyA (GeneArt, Thermo Fisher). The VRV derivative constructs AGG[1], AGG[3] and AGG[1, 3] were also synthesized commercially. Other constructs were prepared by restriction cloning, by using the Q5® site-directed mutagenesis kit (New England Biolabs; NEB), by using the NEBuilder® HiFi DNA assembly master mix (NEB) or with a combination of these methods using the above reporter constructs in the pMK-polyA plasmid as the backbone. Nucleotide sequences of final reporter constructs were verified by Sanger sequencing. Oligonucleotides used for cloning are presented in **S1 Table**.

## *In vitro* transcription

Plasmid DNAs were linearized with HindIII (placed after the polyA tail) and used for *in vitro* transcription using the HiScribe™ T7 high yield RNA synthesis kit (NEB) following the manufacturer's protocol. A cap analog ($m^7G(5')ppp(5')G$; NEB) was used during the transcription reaction (GTP to cap analog 1:4 ratio) for the *in vitro* co-transcriptional capping of ARA transcripts. According to the manufacturer description, 80% of the transcripts will be capped under these conditions. The transcribed RNAs were purified using phenol-chloroform or the Monarch® RNA cleanup kit (NEB). The integrity of the RNAs was verified by agarose gel electrophoresis followed by ethidium bromide staining.

## *In vitro* translation

The reporter transcripts were translated using wheat germ extracts (WGE) *in vitro* translation kit (Promega). The final concentration of potassium acetate was kept at 70 mM and one picomole of transcripts was used in the 50 μl of WGE translation reaction mix (unless otherwise specified). Translation reactions were carried out for 90 minutes in the presence of [35]S Methionine/Cysteine (PerkinElmer; for radiolabeled reaction) or in the presence of non-radiolabeled amino acid mix (for non-labelled reactions). The lysates were then used for luciferase assay or protein visualization.

## Luciferase reporter assay

Luciferase reporter assays were carried out using the Dual-luciferase® reporter assay system (Promega). Ten μl of *in vitro* translated reactions were mixed with 40 μl of 2x passive lysis buffer in a 96 well plate. The luminescence reading for Rluc was recorded after adding 100 μl each of LARII and Stop & Glo reagents (Promega) in a FilterMax F5 Multi-Mode microplate reader (Molecular device).

## Visualization of proteins

Five microliters of the *in vitro* translated reactions were mixed with an equal volume of protein loading buffer (4% SDS, 20% glycerol, 10% BME, 125 mM Tris pH 6.8), boiled for 5 min before loading and separated by SDS-polyacrylamide gel electrophoresis (SDS-PAGE). For the radio-labelled samples, the gels were dried and radioactive bands were visualized using a phosphori-mager (Cyclone Plus, PerkinElmer). For non-radiolabelled samples, the proteins were transferred to a polyvinylidene difluoride membrane (BioRad) and probed with a commercial Rluc antibody (Abcam ab185925). The chemiluminescence was visualized using the Clarity Western ECL blotting substrate (Biorad) and a ChemiDoc XRS (Biorad).

## Results

### Translation of reporter transcripts containing the ToRSV-Rasp1 RNA2 5' region initiates at the first in-frame AUG

The first 900 nts of the 5' region of ToRSV RNA1 and RNA2 are highly identical and include three in-frame AUGs in some isolates, e.g. ToRSV-Rasp1 (**Figs 2A and 2B** and **S1**) [39, 40]. The first in-frame start codon (AUG[1] at nt 77 of ToRSV-Rasp1 RNA2) is conserved in both RNAs and amongst all ToRSV isolates. The second (AUG[2] at nt 146 of ToRSV-Rasp1 RNA2) and third (AUG[3] at nt 440 of ToRSV-Rasp1 RNA2) putative start codons are not conserved in all isolates but could have functional significance when they occur, possibly leading to the production of truncated polyproteins as shown for the RNA2 (M-RNA) of the related cowpea mosaic virus (genus *Como-virus*, family *Secoviridae*) [44–46]. This suggestion is supported by the observation that the third AUG is in a more favourable context for translation initiation (*i.e.*, G or A at -3 and G at +4) [39, 40]. To determine whether translation initiates at the first AUG and/or at one of the two downstream AUG codons, a luciferase reporter transcript (VRV) was used. The 5' end of the VRV transcript corresponded to the first nucleotide of ToRSV-Rasp1 RNA2. The transcript included the 5' region of ToRSV-Rasp1 RNA2 up to the third AUG (nts 1–443). The luciferase ORF was placed immediately downstream of the third AUG codon, preserving the context of this AUG up to the +4 position (AUGG). Translation initiation at the third AUG (AUG[3]) was expected to produce the 34 kDa Rluc protein (black rectangle) (**Fig 3A**). Translation initiation at upstream AUG[1] and AUG[2] would produce fusion proteins that include viral sequences (shown with the grey rectangles) and have calculated molecular masses of 48 and 46 kDa, respectively (**Fig 3A**).

Additional reporter transcripts were also prepared that contained an AUG to AGG mutation introduced in one (AGG[1], AGG[2], AGG[3]), two (AGG[1, 2], AGG[1, 3], AGG[2, 3]) or all three (AGG[1, 2, 3]) putative start codons. This mutation (AUG to AGG) was previously shown to reduce translation of the dihydrofolate reductase gene to 3% of that observed with the wild type (AUG) start codon in the WGE translation system [47]; although background levels of translation initiation from a non-AUG start codon can differ depending on the gene and translation system [48].

The protein translated from the wild-type (WT) VRV transcripts separated on SDS-PAGE with an apparent migration slightly above the 46 kDa marker. Mutation of the first start codon

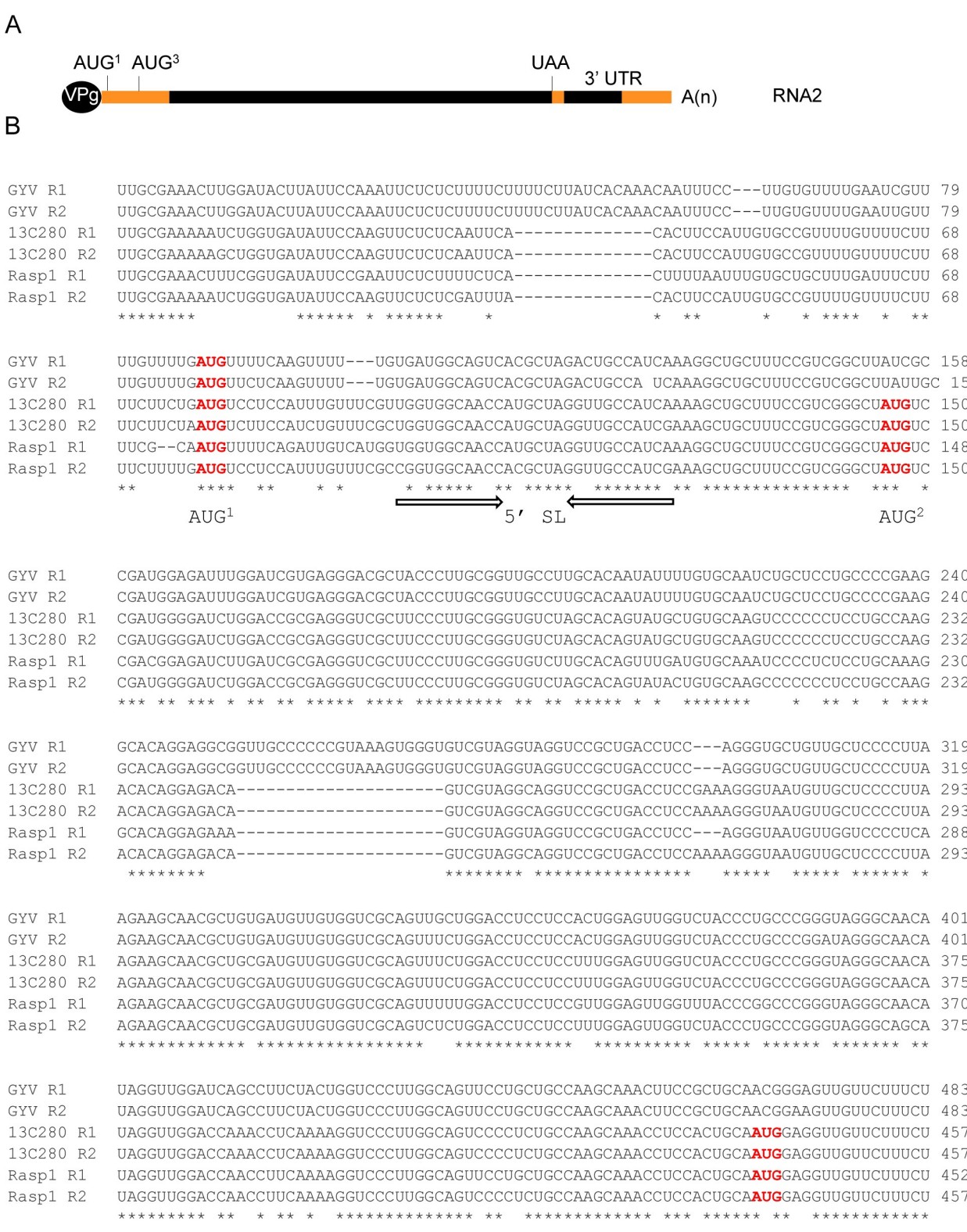

**Fig 2. Regions of sequence identity in the 5' and 3' regions of the genomic RNAs of selected ToRSV isolates.** (A) Schematic diagram of the ToRSV RNA2 showing regions (orange bars) with high sequence identities amongst ToRSV isolates. The 5' region of the two genomic RNAs of many ToRSV isolates are nearly identical in the first 900 nts. The first 150 nts and last 650 nts of the 3' UTRs of the two genomic RNAs are also

highly conserved amongst ToRSV isolates. (B) Sequences alignments of the first 500 nts of the two genomic RNAs (RNA1: R1, RNA2: R2) from selected ToRSV isolates showing putative start codons. The three putative start codons are shown in red. The predicted complementary stem sequences of a putative stem-loop (5' SL) structure located after the first AUG are shown with arrows. NCBI accession number are as follows: ToRSV-13C280 R1 (KM083890), ToRSV-13C280 R2 (KM083891), ToRSV-GYV R1 (KM083892), ToRSV-GYV R2 (KM083893), ToRSV-Rasp1 R1 (KM083894), ToRSV-Rasp1 R2 (KM083895). Please see **S1** and **S2 Figs** for additional sequence alignments of the 5' region and 3' UTR of ToRSV RNAs.

(AGG[1]) resulted in a drastic reduction of the band intensity of the ~46 kDa product, while mutations of the other start codons (AGG[2], AGG[3]) did not (**Fig 3B**), suggesting that the band was produced by translation initiation at the first AUG. Mutation of AUG[1] in mutants AGG[1] and AGG[1, 3] resulted in the appearance of a new slightly faster migrating new band (below the 46 kDa marker). This new product was not observed when AUG[2] was also mutated in AGG[1, 2] or AGG[1, 2, 3] suggesting that translation initiation shifted to AUG[2] when the first AUG is

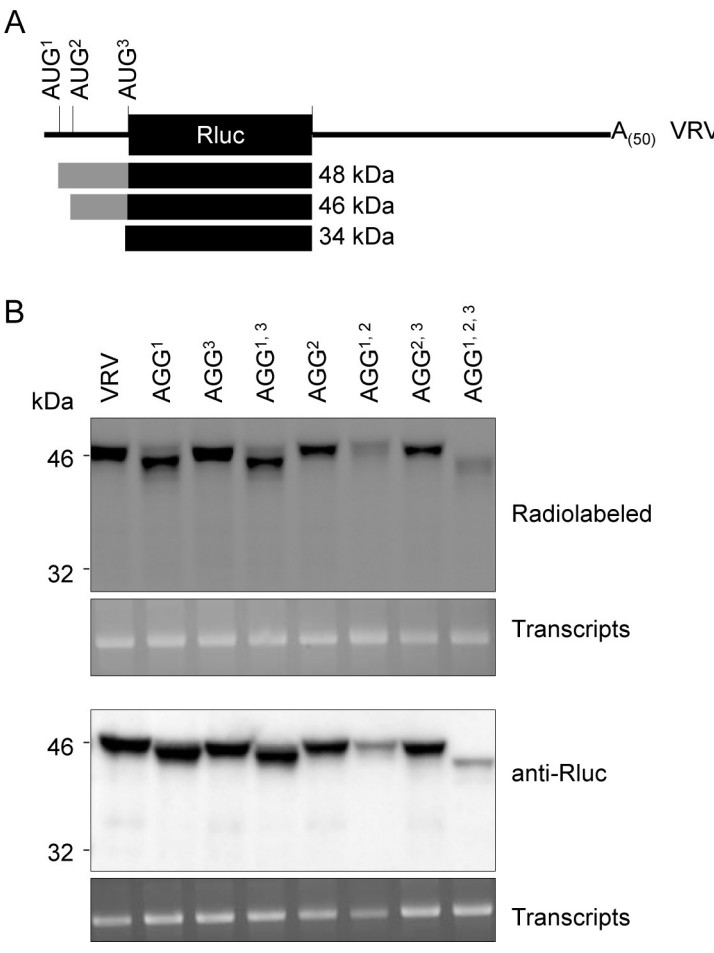

**Fig 3. *In vitro* translation of the VRV transcript initiates at the first AUG of the viral 5' UTR.** (A) Schematic representation of the wild-type reporter transcripts (VRV) with the three putative AUG-start codons (AUG[1], AUG[2], AUG[3]). Fusion Rluc proteins produced from each AUG are shown with rectangles along with their predicted molecular mass. The solid black rectangles represent the Rluc protein and the grey rectangles represent the portion of viral polyprotein sequences fused in-frame with Rluc after translation initiation at AUG[1] or AUG[2]. (B) *In vitro* translation of WT and AUG mutant transcripts. Detection of Rluc by radiolabelling or after probing with Rluc antibody shows the Rluc proteins produced from the wild-type VRV transcript and from the AUG to AGG mutant derivatives. Pictures of EtBr-stained RNA show the integrity of the corresponding transcripts used for the *in vitro* translation. Migration of molecular mass size markers is shown on the left.

mutated. A faint protein band of approximately 44 kDa was observed in the triple mutant AGG[1, 2, 3] and to some extent in mutants AGG[1] and AGG[1, 3] (Fig 3B). The origin of this additional band is not clear. It may have arisen by translation initiation at a non-AUG codon downstream of AUG[2]. It is also interesting to note that while some level of non-canonical translation initiation was observed at or near AGG[1] for the AGG[1], AGG[1,2] and AGG[1,3] mutants, this was not observed for the AGG[1,2,3] mutant. We do not have an explanation for this observation. Importantly, we did not obtain evidence for translation initiation at AUG[3] with any of the transcripts tested, as we did not observe the accumulation of the anticipated 34 kDa protein. Taken together, these results suggested that the first start codon (present at nt 77 of ToRSV-Rasp1 RNA2) is the preferred site of translation initiation *in vitro*, at least in the context of the VRV reporter transcript.

## The 5' region and 3' UTRs from ToRSV-Rasp1 RNA2 are both required for efficient cap-independent translation of the VRV reporter transcript

Before examining the mechanism of VRV translation initiation in more detail, we first performed a titration experiment to determine non-saturating transcript concentrations in our experimental system (S3 Fig). The concentration of one picomole of uncapped VRV transcript per 50μl WGE was shown to be in the linear range of the titration curve and was used for further experiments.

Next, we compared translation of the uncapped VRV transcript to that of a control reporter transcript (ARA) that included the Rluc ORF under the control of the 5' and 3' UTR from the *A. thaliana* actin gene (Fig 4A). As expected, translation of the uncapped ARA transcript only resulted in low levels of luciferase activity (Fig 4B). Capping of ARA (labelled as cARA) prior to translation, enhanced the translation rate as evidenced by the higher level of luciferase activity, which supported the use of canonical cap-dependent translation initiation. Translation enhancement of ARA after capping was modest and was accompanied with a rather large error bar (see cARA transcript, Fig 4B), suggesting that the efficiency of the capping reaction varied from one experiment to another. The luciferase activity produced from uncapped VRV transcripts was significantly enhanced compared to that of the uncapped ARA transcript (Fig 4B). Immunoblots of translation products confirmed that the relative luciferase activities correlated with the relative abundance of the translated proteins (Fig 4C). The enhanced *in vitro* translation of uncapped VRV versus uncapped ARA suggests the presence of translation-enhancing elements in the viral 5' region and/or 3' UTR that enable a cap-independent translation mechanism.

Replacing the viral 3' UTR of the VRV reporter transcript with the actin 3' UTR (mutant VRA) or replacing the viral 5' region with the actin 5' UTR (mutant ARV) resulted in significantly reduced Rluc accumulation compared to the VRV transcript (Fig 4B and 4C), suggesting that neither the 3' UTR nor the 5' region of ToRSV-Rasp1 RNA2 can function alone. Thus, both regions were required for efficient cap-independent translation of the VRV transcripts.

The identification of AUG[1] as the translation initiation site delineated a 76 nts 5' UTR. However, the VRV transcript included sequences in the coding region of the RNA2 polyprotein (between AUG[1] and AUG[3]) which may play a role in translation. To test this, a derivative transcript was prepared that included only the first 76 nucleotides of ToRSV-Rasp1 RNA2 including AUG[1] (V76RV, Fig 4D). *In vitro* translation of V76RV resulted in significantly reduced luciferase activity compared to VRV (Fig 4E). Immunoblotting of the *in vitro* translation products with Rluc antibodies allowed the detection of the expected 48 kDa fusion protein for VRV and the 34 kDa unfused Rluc protein for V76RV, both produced after translation initiation at AUG[1]. The relative accumulation of the unfused Rluc protein produced from

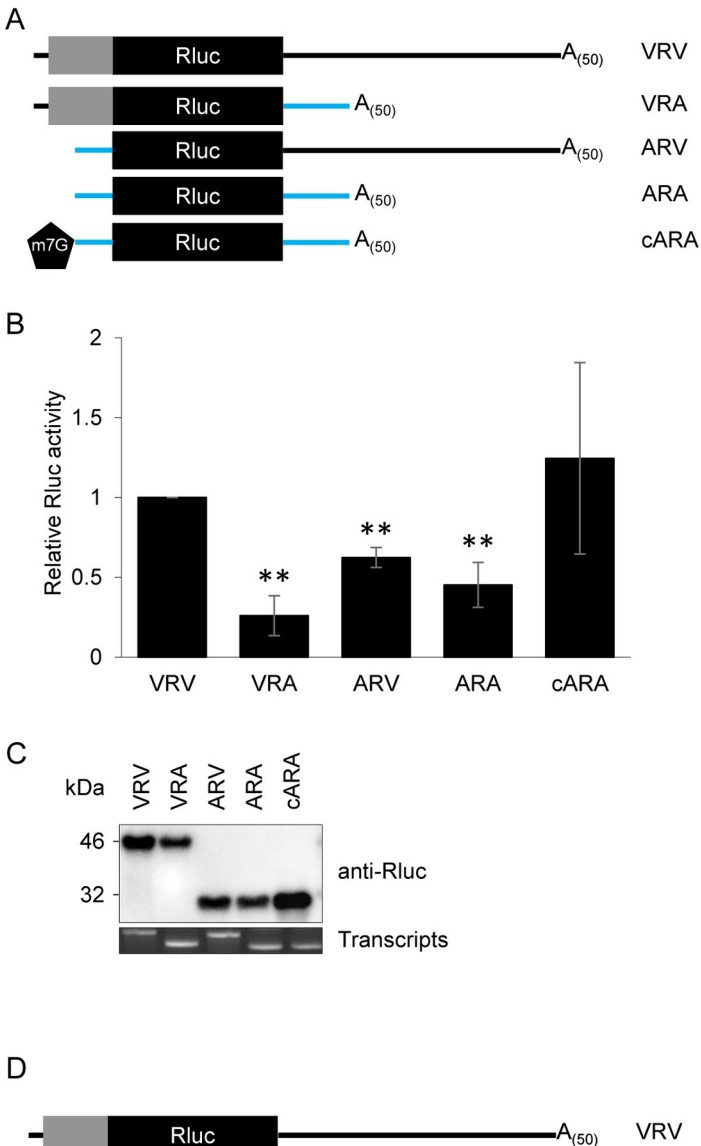

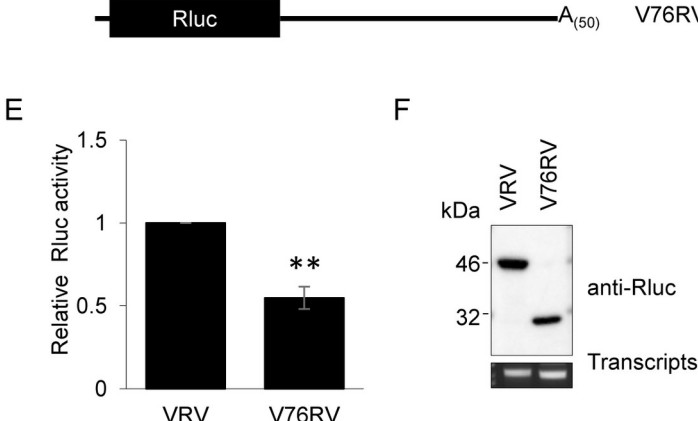

**Fig 4. The ToRSV 5' region and 3' UTR are both required for efficient translation from reporter transcripts.** (A) Schematic representation of VRV and derivative transcripts. In the derivative transcripts, the viral 5' region and/or 3' UTR (black lines) are replaced by the actin 5' and/or 3' UTR (blue lines). Grey rectangles represent the fusion of viral

sequences to Rluc after translation initiation at the first AUG codon. The black pentagon structure represents the 7-methylguanylate cap on the ARA transcript (cARA, for capped ARA) that is enzymatically linked during the *in vitro* transcription reaction. (B) Luciferase activity of *in vitro* translated VRV and derivative transcripts. Rluc activity of the capped ARA (cARA) control transcript is comparable to that of the uncapped VRV transcript (no significant difference between VRV and cARA, P>0.05). Rluc activities of uncapped derivative transcripts missing the 5' region and/or 3' UTR from ToRSV-Rasp1 RNA2 (VRA, ARV and ARA) are significantly lower (**, P<0.001) compared to VRV (Student's t-test, two-tailed, n = 4). Luminescence values obtained after translation of each derivative reporter transcripts were normalized to that obtained for the VRV transcript. Results are shown as the average of four independent experiments, each with three technical repeats. Error bars represent the standard deviation. (C) Protein blot showing the Rluc expression from transcripts shown in B. Proteins were visualized by immunoblotting with an anti-Rluc antibody. EtBr staining of RNA transcripts run on an agarose gel shows the integrity of the transcripts. (D) Schematic representation of VRV and V76RV derivative transcripts. V76RV includes a deletion of the viral polyprotein coding region (deletion from $AUG^1$ up to the nucleotide immediately upstream of $AUG^3$). (E) Luciferase activity of *in vitro* translated VRV and V76RV reporter transcripts. Rluc activity of *in vitro* translated V76RV is significantly lower than that of the VRV transcripts (**, P<0.001) (Student's t-test, two-tailed, n = 5). Luminescence (Rluc) values are normalized to that obtained for the VRV transcripts. Results are shown as the average of five independent experiments, each with three technical repeats. Error bars represent the standard deviation. (F) Protein immunoblot showing the Rluc expression from transcripts shown in E. Pictures of EtBr-stained RNA show the integrity of the transcripts.

V76RV and of the chimeric Rluc derived from the VRV transcripts mirrored the relative luciferase activities measured for the two transcripts (**Fig 4F**). Thus, the inclusion of viral sequences downstream of $AUG^1$ enhanced translation of the VRV transcript.

## Translation of the VRV transcript is inhibited in the presence of m7GpppG cap analog

The addition of excess amounts of free cap analog to *in vitro* translation reactions can reduce the pool of functional eIF4E and inhibit translation of transcripts that are dependent on eIF4E and/or the associated eIF4G [49, 50]. Indeed, translation of the capped reporter transcript (cARA) was clearly affected by the presence of increasing amounts of cap analog (**Fig 5A**), with an average 80% reduction at a 125x molar ratio (**Fig 5B**). Translation of uncapped VRV transcripts was also notably affected by the presence of cap analog (**Fig 5A**), with an average 66% reduction at the same ratio (**Fig 5B**). These results suggest that efficient cap-independent translation of VRV depends on a component(s) of the eIF4F complex.

## Cap-dependent translation of reporter transcripts is inhibited *in trans* by the ToRSV-Rasp1 RNA2 3' UTR

We considered the possibility that the ToRSV-Rasp1 RNA2 3' UTR contains a CITE element that recruits one or several translation factors. When present in excess amounts, CITEs provided *in trans* have been shown to inhibit cap-dependent translation of reporter transcripts by sequestering translation factors [51, 52]. We tested whether the ToRSV-Rasp1 3' UTR contained a CITE-like sequence that could inhibit translation of the control capped ARA transcript (cARA). A transcript corresponding to the entire ToRSV-Rasp1 RNA2 3' UTR (V3', 1547 nts long) was produced and supplied *in trans* at various molar ratios to the cARA transcript. As a control, transcripts derived from the pMK-polyA vector backbone (C, 438 nts long) were also produced and supplied *in trans*. The luciferase assays revealed significant inhibition of cARA translation with increasing concentration of V3' transcripts (**Fig 6A**). When provided at a 20-fold excess molar ratio, the V3' transcript inhibited cap-dependent translation of cARA by approximately 92% (**Fig 6B**). In contrast, translation of cARA was not significantly inhibited with the addition of the control transcript (C) (**Fig 6A and 6B**). This result suggests that the ToRSV-Rasp1 RNA2 3' UTR is sequestering one or several translation factor(s), which are required for cap-dependent translation of cARA.

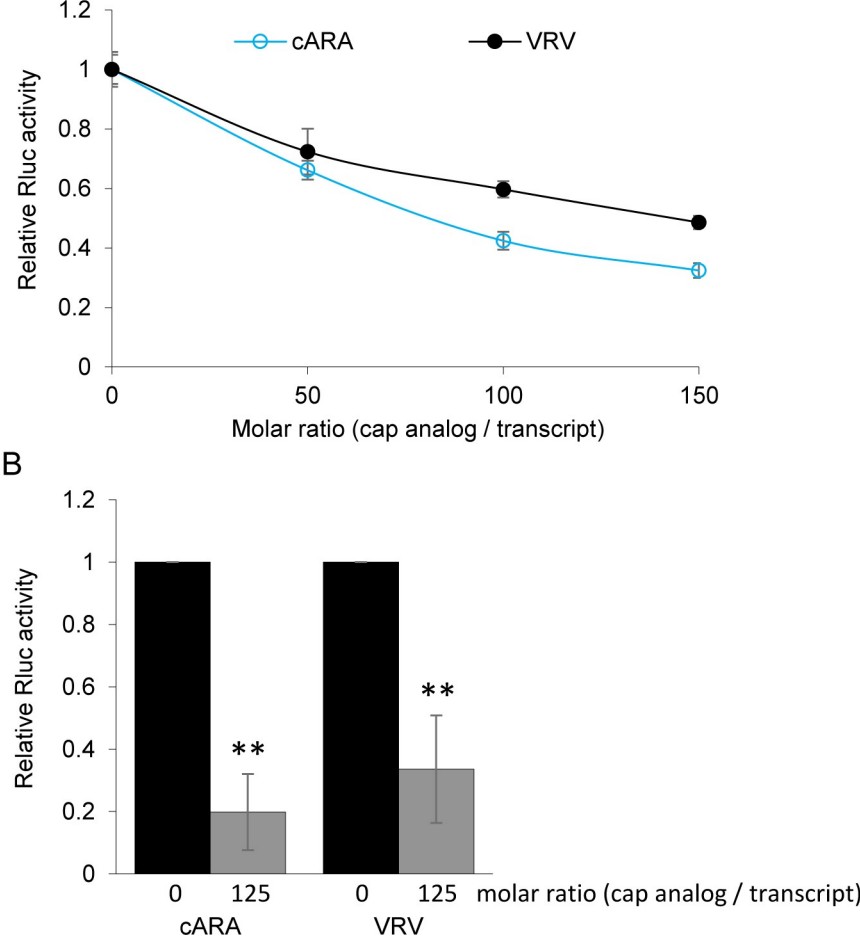

**Fig 5. Cap analog inhibits translation of uncapped VRV transcripts.** (A) Cap analog was added at increasing concentrations ranging from 0 to 150 picomoles per 50 μl translation reaction volume that also contained one picomole of reporter transcripts (uncapped VRV or capped ARA) and the luminescence activity of the translation reactions was measured. Experiments were repeated four times using varying concentrations of cap and a representative result is shown. Error bars represent the standard deviation of three technical repeats. (B) Translation inhibition of uncapped VRV or capped ARA (cARA) transcripts in the presence of a 125-molar excess of cap analog. Reactions were conducted as above. Translation of 1 picomole of either transcript (VRV or cARA) was reduced significantly in the presence of 125 picomoles of cap analog (**, P<0.001, Student's t-test, two-tailed, n = 4). Results are presented as an average of four independent experiments, each with three technical repeats. Error bars represent the standard deviation.

## A 386 nucleotide region of the ToRSV-Rasp1 RNA2 3' UTR is necessary and sufficient for translation of VRV reporter transcripts in conjunction with the viral 5' UTR

All reported viral 3' CITEs consists of relatively short sequences (less than 300 nts) [20, 25]. The ToRSV RNAs 3' UTRs are unusually long (1547 nts for ToRSV-Rasp1 RNA2) and it can be anticipated that some region(s) of the 3' UTR play a more significant role in translation than others. VRV derivative transcripts were prepared that contained non-overlapping deletions of the viral 3' UTR (**Fig 7A**). The 3' UTR was arbitrarily divided into four regions (1–4) and each region was deleted to produce VRVΔ1 (deletion of the first 386 nts; nts 6017–6402 of RNA2), VRVΔ2 (deletion of 386 nts; nts 6403–6788), VRVΔ3 (deletion of 386 nts; nts 6789–

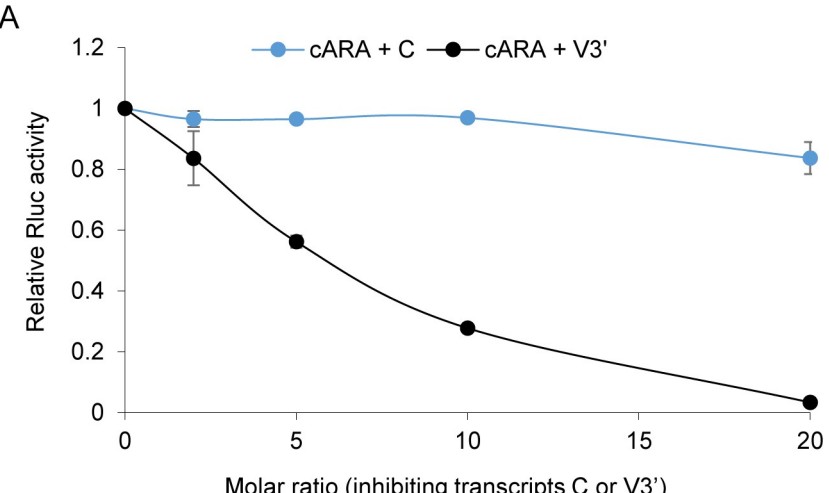

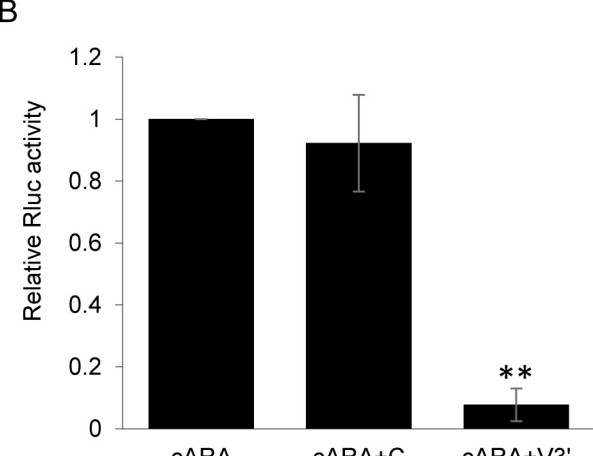

**Fig 6. Translation of capped ARA control transcripts is inhibited by the viral 3' UTR supplied *in trans*.** (A) Luciferase assay showing translation inhibition of capped reporter transcripts (cARA) with increasing concentration of viral 3' UTR (V3') supplied *in trans*. Reporter transcript (cARA) was used at a concentration of one picomole per 50 μl of WGE translation reactions. Either a control transcript (C, derived from vector sequences) or the V3' transcript were supplied at various concentrations (0, 2, 5, 10 and 20 picomoles per 50 μl of translation reactions) and the luminescence was measured after *in vitro* translation. The luminescence values of reporter transcripts with added inhibitory transcripts (C or V3') were normalized to those obtained for the cARA in the absence of inhibitory transcripts. Results are shown as the average values obtained from two independent experiments, each with three technical repeats. Error bars represent the standard deviation. (B) Inhibition of translation of cARA transcripts in the presence of a 20-fold excess of control or viral 3' UTR provided *in trans*. Translation reactions were programmed as in A with 1 picomole of cARA transcripts and 20 picomoles of either V3' or C transcripts. Translation of cARA was reduced significantly (\*\*, P <0.001) in the presence of 20 picomoles of V3' transcripts (Student's t-test, two-tailed, n = 3). Results are shown as an average of three independent experiments, each with three technical repeats. Error bars indicate the standard deviation.

7174) and VRVΔ4 (deletion of 389 nts; nts 7175–7563) (**Fig 7A**). Deletion of region 3 (VRVΔ3) significantly reduced translation efficiency (>75% reduction compared to WT VRV) while other deletions did not (**Fig 7B**). This suggested that region 3 (nts 6789–7174) is required for efficient translation. To test whether this region is sufficient for translation, another VRV derivative was produced that contained only region 3 (VRVΔ124) (**Fig 7A**). This transcript translated to a level similar to that of the VRV transcript suggesting that region 3 of

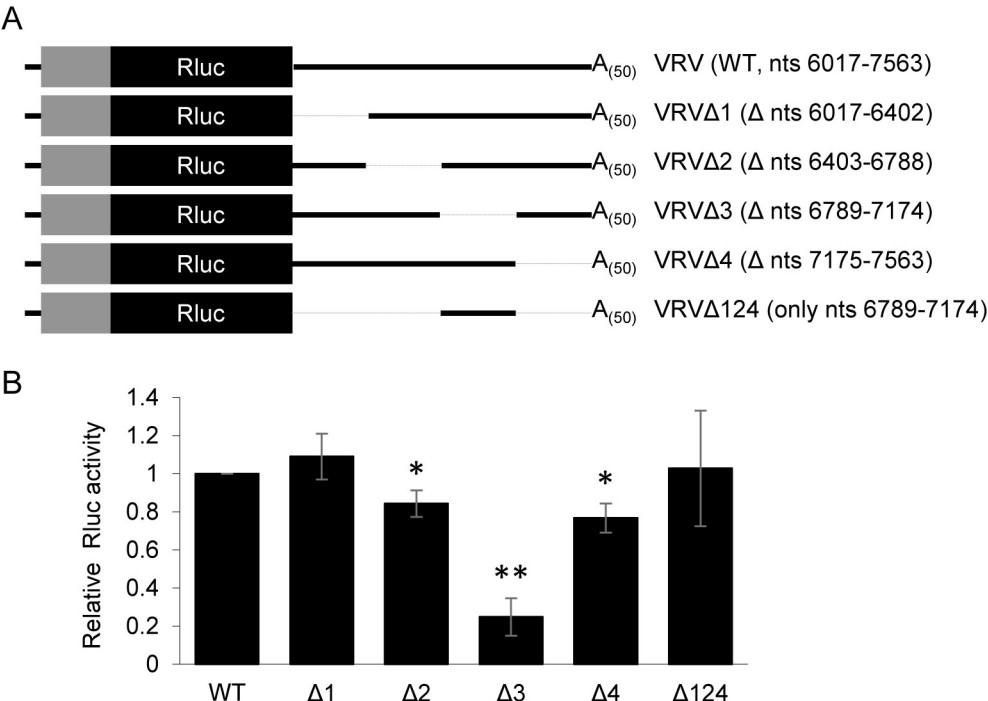

**Fig 7. A 386 nucleotides region of the viral 3' UTR is necessary and sufficient for translation of reporter transcripts in conjunction with the viral 5' region.** (A) Graphical representation of VRV derivative transcripts containing non-overlapping deletions of ~400 nts regions in the 3' UTR. Numbers in parentheses indicate the region of the ToRSV-Rasp1 RNA2 3' UTR included in the construct (WT and Δ124) or the regions that were deleted from constructs Δ1 to Δ4 (numbering from the 5' end of RNA2). (B) Luminescence activity from the *in vitro* translated VRV and deletion derivative transcripts. Luciferase activities obtained from the deletion derivative transcripts VRVΔ1 and VRVΔ124 are comparable (P>0.05) to that of the WT VRV transcript. Those of VRVΔ2 and VRVΔ4 are only slightly reduced (25% and 33% reduction, respectively, *, P<0.05) compared to WT VRV. In contrast, the luminescence activity obtained from the VRVΔ3 transcript is significantly reduced (>70% reduction, **, P<0.001) compared to WT VRV (Student's t-test, two-tailed, n = 3). The luminescence values from the derivative transcripts are normalized to the value obtained from VRV transcripts and the averages of three independent experiments, each with three technical repeats, are shown. Error bars represent the standard deviation.

the 3' UTR is both necessary and sufficient for cap-independent translation together with the 5' UTR (**Fig 7B**).

Derivatives reporter transcripts containing small non-overlapping deletions of region 3 were prepared (**Fig 8A**): VRVΔ3a (deletion of 100 nts; nts 6789–6888), VRVΔ3b (deletion of 100 nts; nts 6889–6988), VRVΔ3c (deletion of 100 nts; nts 6989–7088) and VRVΔ3d (deletion of 86 nts; nts 7089–7174). Transcripts VRVΔ3a and VRVΔ3b exhibited a significant reduction of translation rate (~70% reduction compared to the VRV transcript) that was comparable to the translation rate observed for VRVΔ3 (**Fig 8B**). These results suggest that sequences contained within nts 6789–6988 of the ToRSV-Rasp1 RNA2 3' UTR are vital for efficient cap-independent translation of the VRV transcript.

## Discussion

### Translation initiation at the first AUG and lack of evidence for leaky scanning in spite of a poor codon context

Using reporter transcripts containing the first 443 nts of ToRSV-Rasp1 RNA2 and including the first three in-frame AUG codons, we show that *in vitro* cap-independent translation of

A

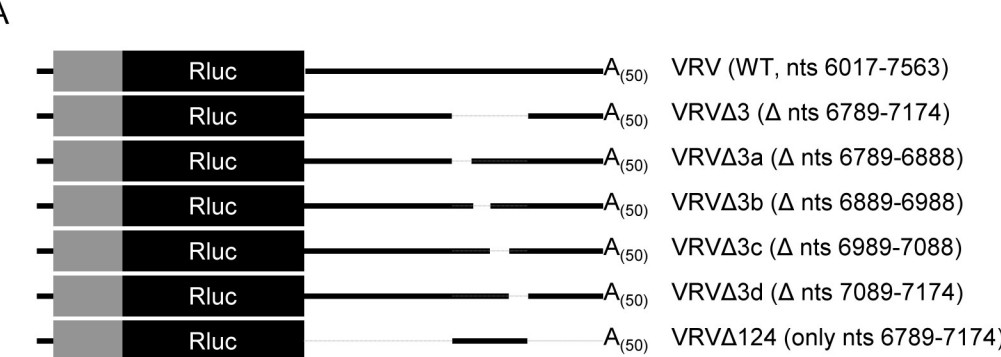

B

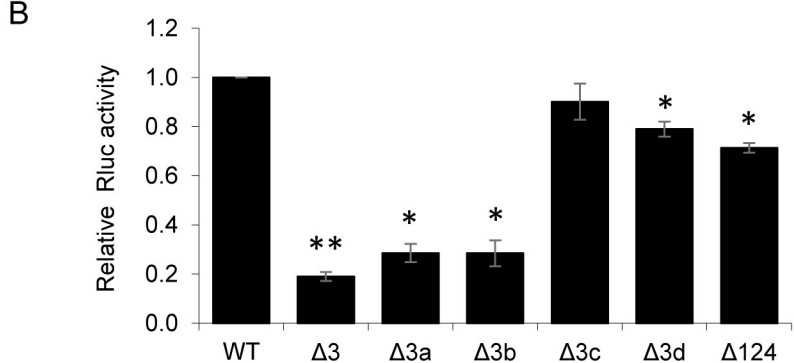

**Fig 8. Delineation of sequences required for translation of uncapped VRV transcripts in region 3 of the 3' UTR.** (A) Graphical representation of VRV derivative transcripts containing non-overlapping 100 nts deletions in region 3 of the 3' UTR. Numbers in parentheses indicate the region of the ToRSV-Rasp1 RNA2 3' UTR included in the construct (WT and Δ124) or the regions that were deleted from constructs Δ3 and Δ3a to Δ3d (numbering from the 5' end of RNA2). (B) Luminescence values from the *in vitro* translated VRV and deletion derivative transcripts. As shown in Fig 7, deletion of the entire region 3 (Δ3) resulted in a significant reduction of translation activity (**, <0.001) (Student's t-test, two-tailed, n = 2). Translation of the VRV reporter transcript was also significantly decreased upon deletion of the first two 100 nts stretches (Δ3a and Δ3b) in region 3 of the 3' UTR (*, P<0.05) (Student's t-test, two-tailed, n = 2). The luminescence values from the derivative transcripts are normalized to the value obtained from VRV transcripts and the average of two independent experiments, each with three technical repeats, are shown. Error bars represent the standard deviation.

these transcripts initiated at the first AUG (AUG[1]) (**Fig 3**). The consensus codon context for higher plants is (A/G)(A/C)aAUGGC with the G at the +4 position and the purine at the -3 position being the most conserved [17–19]. According to this consensus, AUG[1] (UUG**AU-GUC**) and AUG[2] (**G**CU**AUG**UC) are predicted to be in a poor context, while AUG[3] (**GCAAUGG**A) would be in a better context. Even though AUG[1] is in a sub-optimal context, we did not obtain evidence for leaky scanning and translation initiation at the more favorable AUG[3]. Three lines of evidence supports this conclusion: (a) we did not observe proteins initiated at AUG[2] or AUG[3] when AUG[1] was present (e.g., in the wild-type transcript), (b) after mutation of AUG[1], translation initiation shifted to AUG[2] (confirming its activity) and (c) mutation of both AUG[1] and AUG[2] reduced cap-dependent translation to very low levels, with proteins initiated at a position close to the mutated AGG[1], but not at AUG[3] (**Fig 3**). It is well-documented that viral RNAs that present their first AUG in a weak codon context can direct the production of alternate proteins initiated at downstream AUGs by allowing leaky scanning of the ribosomes [3]. For example, two polyproteins with alternate N-terminal ends are produced from cowpea mosaic virus RNA2, and this has been attributed to leaky scanning [44–

46]. Thus our results raise the question: what prevented ribosome leaky scanning from the sub-optimal AUG[1] in the ToRSV-Rasp1 RNA2 reporter transcript?

It has been suggested that translation initiation at AUGs in sub-optimal contexts can be enhanced when scanning 40S ribosomal subunits are temporarily stalled by stable RNA stem-loop structures naturally present or artificially inserted 14–15 nts downstream of sub-optimal start codons [12, 53, 54]. There are precedents for the role of secondary structures in the coding regions of viral RNAs as enhancers of translation accuracy. For example, translation initiation from the first AUG of Dengue virus RNA, which is in a sub-optimal initiation context, is facilitated by a stem-loop structure located 14 nts downstream [55]. Stable secondary structures located further downstream can also influence the start codon selection [56, 57]. Indeed, a stable hairpin structure (with Gibbs free energy $\Delta G$ = -76.90 kcal/mol) located 27 nts downstream of the start codon was shown to enhance translation of Sindbis virus subgenomic RNA and prevent leaky scanning, even when the AUG was mutated to CUG [56]. We searched for possible secondary structures downstream of AUG[1] in ToRSV-Rasp1 RNA2. While the 5' UTR of ToRSV RNA2 (defined as the 76 nts upstream of AUG[1]) is AU-rich, the 361 nts region between AUG[1] and AUG[3] has a higher GC content (>50%). *In silico* secondary structure predictions revealed several putative stem-loop structures in this region. In particular, a stable stem-loop structure (5' SL, $\Delta G$ = -19 kcal/mol) was predicted 17 nts downstream of AUG[1] (**Fig 9A and 9B**). Nucleotide sequence alignment of ToRSV isolates identified base pair covariation that preserved the predicted 5' SL structure (**Fig 9B and sequences indicated by the two arrows in Fig 2B**), suggesting that it has a functional role in translation or in another step of the viral replication cycle. Indeed, the reduced level of translation observed for V76RV, in which the 5' proximal coding sequence (located 3' of AUG[1] and including the 5' SL) was deleted, is consistent with this concept (**Fig 4D and 4E**). However, further experiments will be necessary to determine if the predicted 5' SL and/or other sequences/structures in the coding region of ToRSV-Rasp1 RNA2 influence the start codon selection and help prevent leaky scanning.

We examined the context of putative start codons for other nepoviruses (**S2 Table**). Although some nepovirus RNAs (CLRV RNAs, BRV RNA1, RRSV RNA1, GBLV RNA1) present their first AUG in a conserved context predicted to be favorable for translation initiation, many have their first AUG in a poor context but incorporate a predicted stable stem-loop structure similar to that suggested for ToRSV RNAs (BRV RNA2, GBLV RNA2, ArMV and GFLV RNAs, GDefV RNA1, PBRSV RNA1, CNSV RNA1). In a third class of nepovirus RNAs, the first AUG is presented in a poor context but is not followed by a predicted stable stem-loop (BLSV RNA2, SLSV RNA1, TRSV RNA2, MMLRaV RNA2, GDefV RNA2, MMMoV RNAs, AILV RNA1, TBRV RNA1). In those cases, it is possible that leaky scanning leads to the formation of alternate polyproteins or that other mechanisms enhance translation at the first AUG. This analysis suggests that regulation of translation initiation is likely to be diverse in nepoviruses and that authentic translation initiation sites should be experimentally determined. Similar variations in start codon choices are observed in other virus genera, for example in the genus *Potyvirus* [24]. While translation of many potyvirus RNAs initiates at the first AUG, plum pox virus RNA shows evidence of leaky scanning with translation initiation preferred at the second AUG [60].

## Contributions of the 5' region and 3' UTR of ToRSV-Rasp1 RNA2 to cap-independent translation of VRV transcripts

The majority of viral IRESs and CITEs function by recruiting eIF4E and/or eIF4G (in plants also eIF(iso)4E and/or eIF(iso)4G), although some can bind ribosome subunits in the absence

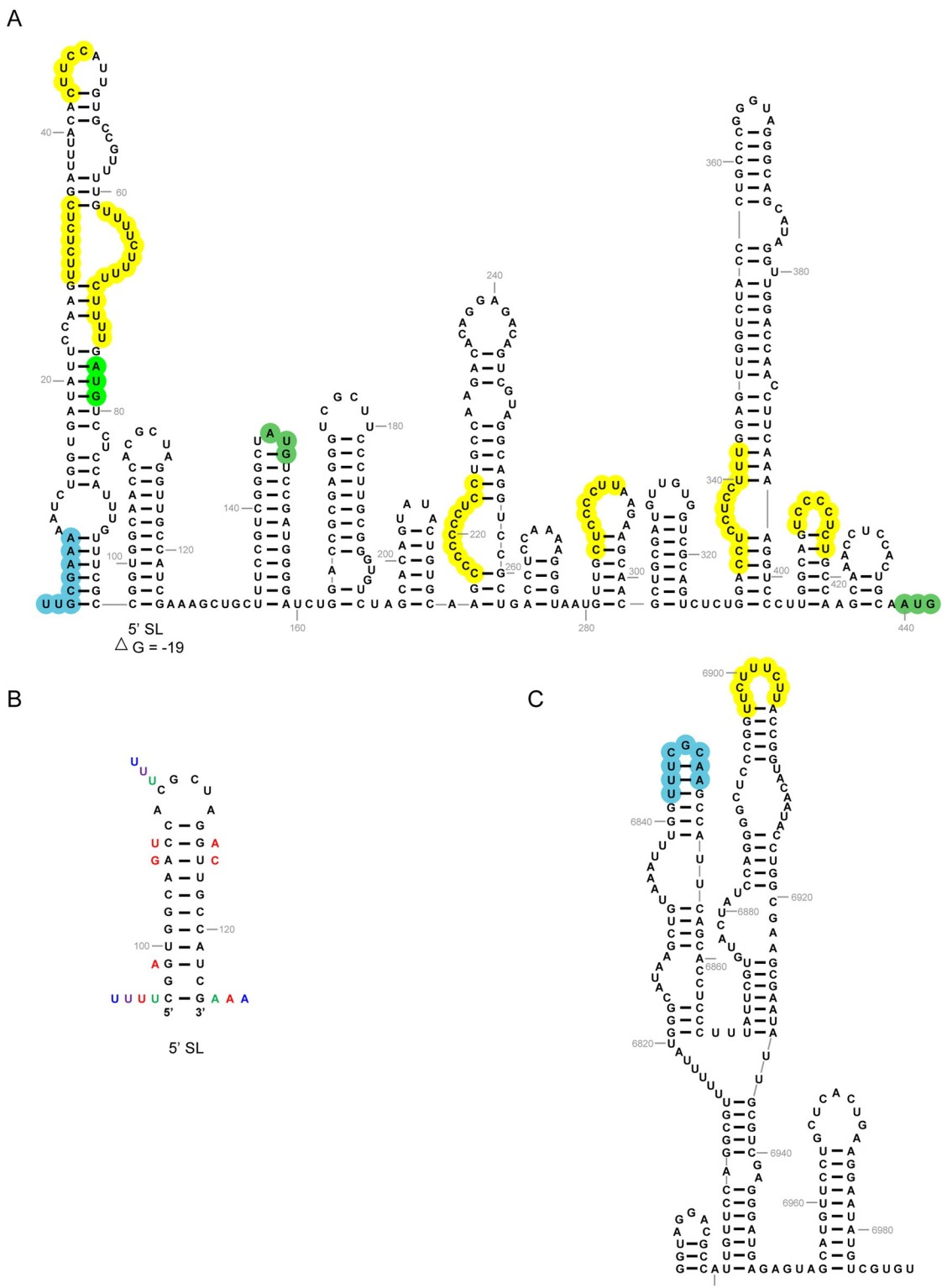

**Fig 9. Predicted secondary structures of the 5' region of ToRSV-Rasp1 RNA2 and of region 3a-b of the 3' UTR.** (A) Predicted secondary structure of the 5' region (first 443 nts up to the third AUG codon) of ToRSV-Rasp1 RNA2. The three AUG codons are shown in green. The predicted 5' SL located 17 nts downstream of the first AUG is shown along with the predicted free energy. A putative base pairing (sequence complementarity) with a sequence from region 3a-b of the 3' UTR shown in panel C is also indicated with the light blue highlight (B) Close-up of the 5' SL structure. Sequence variation occurring in ToRSV isolates are shown

with the following color code: ToRSV-Rasp1 RNA2 (black), ToRSV-Rasp1 RNA1 (green), ToRSV-GYV RNA1 and RNA2 (red), ToRSV-13C280 RNA2 (purple), ToRSV-13C280 RNA1 (blue). Please note that complementarity of sequences in the stem is conserved for all isolates. Please see sequence alignment of the corresponding region in **Fig 2**. (C) Predicted secondary structure of region 3a-b (nts 6789–6988) of the 3' UTR of ToRSV-Rasp1 RNA2. A putative base pairing (sequence complementarity) with a sequence from the 5' region shown in panel A is indicated in light blue. For all panels, secondary structures were predicted using the mFold webserver [58]. The obtained dot-bracket format files were then loaded into RNA2Drawer (web application) [59] for drawing and visualization. Polypyrimidine stretches in exposed loops or bulges in the predicted structures are highlighted in yellow. Please see **S4 Fig** for secondary structure predictions of the same RNA regions for other selected ToRSV isolates.

of translation initiation factors as exemplified by the intergenic IRES of viruses in the family *Dicistroviridae* [21] or the 3' CITE (T-shaped structure) of turnip crinkle virus [28]. We show that *in vitro* translation of uncapped VRV transcript was significantly impaired in the presence of excess cap analogue (**Fig 5**), indicating that components of the eIF4F and/or eIF(iso)4F cap-binding complexes are necessary for the cap-independent translation of VRV transcripts.

Using deletion mutagenesis, we obtained evidence for *cis*-acting translation enhancing elements in the 5' region and 3' UTR of ToRSV-Rasp1 RNA2. Independently, neither of these elements could function efficiently (**Fig 4**), suggesting that a cross-talk between the 5' and 3' ends of the RNAs occurs either directly through long distance RNA-RNA interactions or indirectly through the formation of a protein bridge. Non-overlapping deletion mutagenesis allowed for preliminary mapping of these *cis*-acting elements. Substitution of the RNA 5' region (including the 5' UTR and proximal coding region, ARV transcript, **Fig 4B**) or deletion of the 5' coding region (V76RV transcript, **Fig 4E**) resulted in an approximately 2-fold reduction of translation efficiency. Substitution of the entire 3' UTR (VRA transcript, **Fig 4**) or deletion of the 386 nts region 3 of this UTR (VRVΔ3, nts 6789–7174, **Fig 7**) reduced the translation rate by approximately 4-fold. Interestingly, deletions of other regions of the 3' UTR did not impact translation efficiency (VRVΔ124, **Fig 7**), suggesting that region 3 of the 3' UTR is necessary and sufficient for cap-independent translation in conjunction with one or several elements in the 5' region of the genome. The presence of a CITE-like element in the 3' UTR was also supported by competition experiments that showed the inhibition of a capped reporter transcript in the presence of excess amounts of the 3' UTR provided *in trans* (**Fig 6**). Further mapping narrowed down the *cis*-acting element to a 200 nts segment of the 3' UTR (region 3a-b, **Fig 8**).

Viral 3' CITEs are normally highly structured and many characterized CITEs have been shown to promote RNA circularization through kissing-loop interactions with stem-loops in the 5' UTR [20, 25, 61]. There are also documented examples of interactions between 3' CITEs and stem-loops present in the proximal 5' coding region, as demonstrated for pea enation mosaic virus [62, 63]. In other cases, for example turnip crinkle virus, kissing loop interactions between the 5' and 3' ends of the RNA could not be identified. Instead, it was suggested that RNA circularization is promoted by protein bridges mediated by ribosome subunits binding separately to the 3' CITE and to a pyrimidine-rich sequence in the 5' UTR [29]. We considered possible secondary structures of region 3a-b in ToRSV-Rasp1 RNA2, which has a GC content of 50%, and examined possible long-distance interactions with the 5' region of the RNA (**Fig 9C**). A putative branched structure with two stem-loops was predicted in region 3a-b. This structure was also predicted for the RNA1 of ToRSV-Rasp1 and the two RNAs of ToRSV-13C280 but not for the two RNAs of the divergent ToRSV-GYV isolate (**S4 Fig**). We could not identify obvious kissing loop interactions between region 3a-b and the 5' region of ToRSV--Rasp1 RNA2. A possible base pairing was detected between the loop sequence of the 5' side hairpin of the putative branched structure in region 3a-b of ToRSV-Rasp1 RNA2 and a complementary sequence at the 5' end of the RNA (highlighted in light blue, **Fig 9A and 9C**). Similar regions of sequence complementarity with the 5' end of the RNA were also found in

predicted loops in region 3a-b of the RNAs of other ToRSV isolates (**S4 Fig**). The sequence at the 5' end of the RNA (UUGCGAAA) is strictly conserved in all ToRSV isolates (**Fig 2B**) and is located at the base of a long hairpin in the secondary structure model (**Figs 9A** and **S4**). Because the hairpin is interrupted by several large bulges and includes a majority of A:U and U:G base pairs, the validity of the predicted secondary structure is uncertain. Whether the hairpin is formed or not would determine the accessibility of the UUGCGAAA sequence for base-pairing with region 3a-b. It is possible that the proposed interaction between the RNA 5' end and region 3a-b promotes ToRSV-Rasp1 RNA2 circularization to enhance translation. Alternatively, this predicted base pairing may only occur under certain conditions (e.g., after disruption of the long hairpin by scanning ribosomes initiating translation at AUG[1]) and could act as a riboswitch to regulate the activity of the 3' CITE. The reduced translation activity of V76RV, which would eliminate base-pairing at the base of the long hairpin, would be consistent with this alternative model. An additional possible base-pairing between region 3a-b and the 5' region was identified in exposed sequences (loops or bulges) in the structure model but was not consistently predicted for all ToRSV isolates (**S4 Fig**). The validity and biological significance of the secondary structures and of the regions of sequence complementarity predicted for ToRSV-Rasp1 RNA2 will need to be tested experimentally.

Several polypyrimidine stretches were previously identified in the ToRSV RNA2 5' UTR region [43]. These regions map to predicted exposed regions of the implied secondary structures (loops or stem bulges) (**Fig 9A**). We found additional polypyrimidine stretches in other predicted loop and stem bulges in the proximal 5' coding region and in region 3a-b of the 3' UTR (**Fig 9**). Most of these polypyrimidine stretches are conserved in the RNAs of other ToRSV isolates (**S4 Fig**) and may interact with ribosome subunits (through base-pairing with the 18S rRNA) or to polypyrimidine tract binding proteins, as previously demonstrated for other *cis*-acting translation enhancing sequences of plant and mammalian viruses [29, 64, 65].

Although the VRV transcripts allowed the identification of translation-enhancing elements in the 5' region and 3' UTR of ToRSV-Rasp1 RNA2, they differ from viral RNAs in that they are missing large sections of the polyprotein coding region. Therefore, we cannot exclude the possibility that translation of RNA2 could be further enhanced by *cis*-acting elements present in other sections of the polyprotein coding region. The VRV transcripts are also lacking the 5' terminal covalently linked VPg. The VPg of nepoviruses (2–3 kDa) is similar to that of picornaviruses [31] and much smaller than the potyvirid VPgs (~20 kDa), which are known to interact with eIF4E (and/or eIFiso4E) [66, 67] and to enhance viral RNA translation *in cis* (as shown for potato virus Y) [68] or *in trans* (as shown for potato virus A) [69]. In contrast, the VPg of picornavirus RNAs is removed from the viral RNA prior to translation and does not play a major role in translation [70, 71]. The role of VPg in nepovirus RNA translation has not been studied in details and it is not known whether or not translating nepovirus RNAs are covalently linked to VPg. In fact, early studies showed that removal of VPg by proteinase K treatment does not affect *in vitro* translation of nepovirus RNAs [72–74]. The ToRSV VPg-protease intermediate polyprotein interacts with eIF(iso)4E, at least *in vitro* [75]. The interaction was mapped to the protease domain, although presence of the VPg domain was shown to enhance the interaction. Although the VPg-protease-eIF(iso)4E interaction may contribute to the translation of viral RNAs *in trans*, this remains to be determined experimentally.

## Translation mechanisms of ToRSV and BRV RNAs: Similarities and differences

While nepovirus 3' UTRs are highly conserved amongst isolates of the same species, they are very divergent from one species to another with their length varying from 1300–1600 nts for

subgroup C nepoviruses to 200–400 nts for subgroup A-B nepoviruses [31, 32]. Phylogenetic analysis using the nucleotide sequence of the larger subgroup C 3' UTRs confirms their diversity (S5 Fig). The 3' UTRs of ToRSV isolates group in a separate branch in this analysis and are distinct from those of other nepoviruses. The 5' regions of nepovirus RNAs are also divergent although they share the common characteristic of including several polypyrimidine stretches [43]. The only other nepovirus for which translation mechanisms have been characterized is BRV [41–43]. Although both subgroup C nepoviruses, the 3' UTRs of ToRSV-Rasp1 RNA2 (1547 nts in length) and BRV RNA2 (1363 nts in length) share only 32% nt sequence identity. The 5' UTRs also vary in their lengths: 76 nts for ToRSV RNA2, 75 nts for ToRSV RNA1, 161 nts for BRV RNA2 and 66 nts for BRV RNA1, assuming translation initiation at the first AUG. Thus, it is perhaps not surprising that although ToRSV and BRV RNAs share similar general translation mechanisms, differences were observed in the positioning, sequence and putative structure of translation-enhancing elements.

That cap-independent translation of ToRSV-Rasp1 RNA2 requires both the 5' region and 3' UTR is similar to previous observations with BRV RNAs [41–43]. In BRV RNAs, a CITE was mapped to the proximal region of the 3' UTRs [41, 42] (region A2, Fig 10A). Deletion of this region reduced the translation efficiency by 84% in BRV RNA2 reporter transcripts [42]. The authors provided evidence for a long-distance kissing loop interaction between predicted stem-loops, one located immediately downstream of the stop codon (3' SL-1) and one in the 5' UTR (5' SL) (highlighted in light blue in Fig 10C). Mutation of the regions of sequence complementarity in 5' SL or in 3' SL-1 reduced the translation rate by 82–96%. Restoring the base pairing by compensatory mutations introduced in the two stem-loops resulted in enhanced translation rate (177%) compared to the wild-type construct [42]. A similar kissing-loop interaction, although with different sequences, was identified between predicted stem-loops in the 5' UTR and the proximal region of the 3' UTR of BRV RNA1 [41]. Karetnikov and colleagues predicted base-pairing between the 5' UTR and the proximal 3' UTR of other nepovirus RNAs, including ToRSV RNAs [42]. However, our results do not support a translation activity for the proximal 3' UTR of ToRSV-Rasp1 RNA2. Indeed, deletion of this region of the 3' UTR did not significantly affect translation efficiency (transcript VRVΔ1 and VRVΔ124, Fig 7). Instead, our results highlight the translation-enhancing activity of the distal region 3a-b of ToRSV-Rasp1 RNA2 3' UTR (Figs 8 and 10A). The corresponding region of BRV RNA2 (region C2) was shown to have modest translation-enhancing activity, as deletion of this region reduced translation by 50% [42] (Fig 10A). A putative kissing-loop interaction was proposed between region C2 of BRV RNA2 and the 5' SL (Figs 10A and S6), although deletion of the putative base pairing sequence in region C2 only had a modest effect on translation of BRV RNA2 reporter transcripts [42]. In addition, deletion of the corresponding region of the 3' UTR did not affect translation of BRV RNA1 [41], even though the 3' UTRs of BRV RNA1 and RNA2 are highly conserved (94.8% sequence identity) [76].

Comparing the 5' regions of ToRSV and BRV RNA2 reveals similar features, including the presence of polypyrimidine stretches and putative long hairpins that encompass the first AUG (Fig 10B and 10C). As discussed above, in both cases the first AUG is followed by a hairpin (labelled 5' SL for ToRSV and 5' SL-2 for BRV, Fig 10B and 10C) that may assist in the selection of this AUG as the site of translation initiation. In the case of the longer BRV RNA2 5' UTR, an additional hairpin (labelled 5' SL, Fig 10C) is located near the RNA 5' end and is critical for promoting translation via a long-distance interaction with the proximal 3' UTR (region A2). While a similar hairpin was not predicted in ToRSV RNAs, regions of sequence complementarity between the RNA 5' end and the distal 3' UTR (region 3a-b) were identified, although the biological significance of this interaction remains to be determined.

Although previous studies with BRV RNAs also took advantage of luciferase reporter transcripts to quantify translation activity, the BRV reporter transcripts only included the 5' and 3'

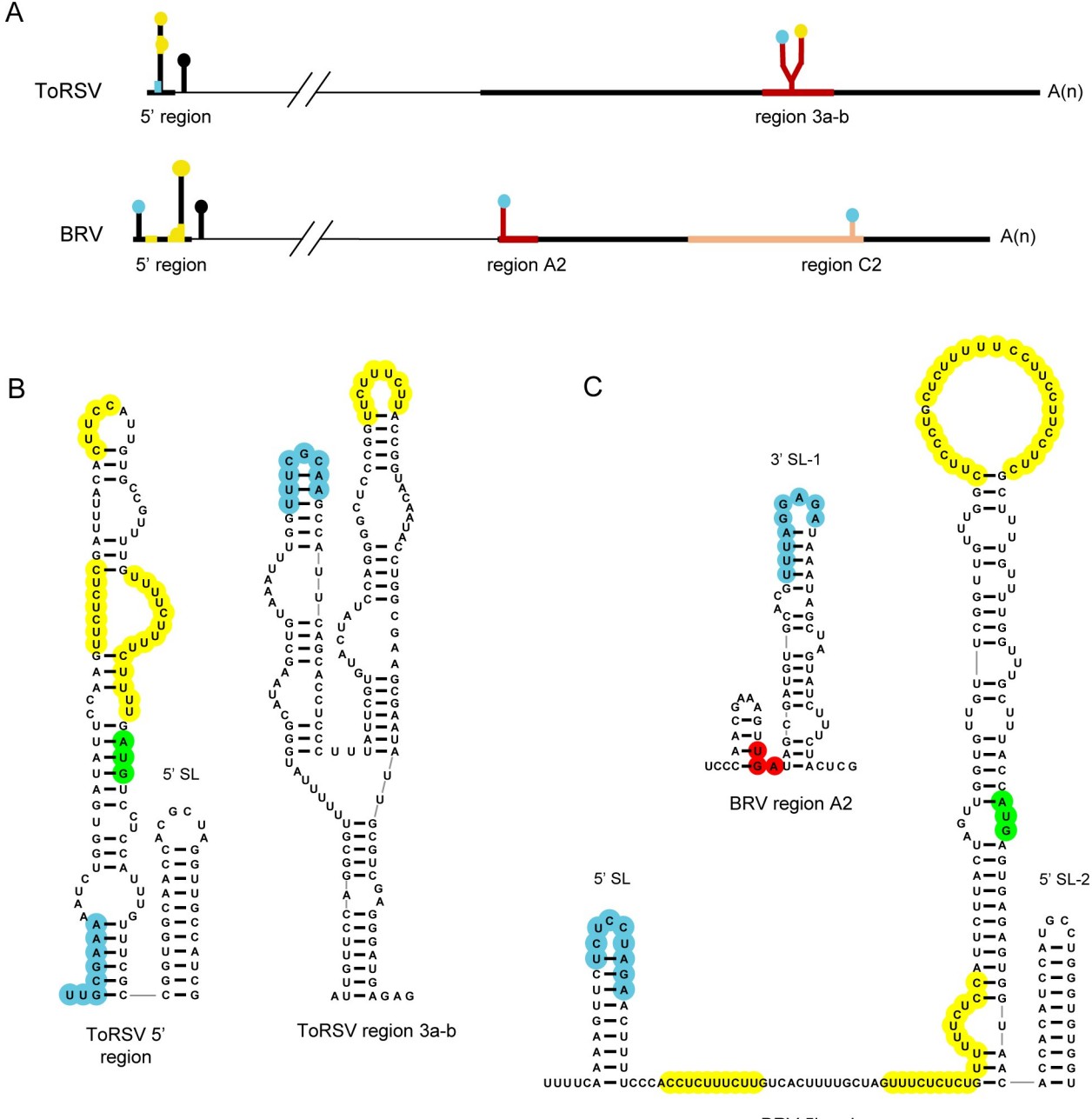

**Fig 10. Comparison of identified translation-enhancing elements and predicted secondary structures in BRV and ToRSV-Rasp1 RNA2s.** (A) Schematic representation of the location of translation-enhancing elements. The RNA2s of ToRSV and BRV are shown with the coding region of the polyprotein represented by thin horizontal black lines and the UTRs by thicker horizontal black lines. Regions in the 3' UTRs identified as playing a critical role in translation are shown in red. A region of the BRV 3' UTR identified as playing a modest role in translation is shown in orange. (B) Predicted secondary structures of the ToRSV-Rasp1 RNA2 5' region and 3' UTR region 3a-b. Structures discussed in the text are shown. For a more detailed representation of ToRSV RNAs features please refer to **Figs 9** and **S4**. (C) Predicted secondary structures of the BRV RNA2 5' region and 3' UTR region A2. Structures discussed in the text are shown. For a more detailed representation of BRV RNA2 features (including a predicted stem-loop in region C2), please refer to **S6 Fig**. For all panels, identified regions of sequence complementarities between the 5' and 3' UTRs are shown in light blue. Polypyrimidine stretches are shown in yellow. The first AUG and the stop codon are highlighted in green and red, respectively.

UTRs from the viral RNAs and did not incorporate viral coding sequences [41–43]. The experimental system also differed: *N. benthamiana* protoplasts for the BRV studies vs the high-fidelity wheat germ extracts *in vitro* system used in this study. We acknowledge that these differences in experimental set-ups could have influenced the activity of translation-enhancing elements. However, it is also possible (and perhaps more likely) that the noted differences in the position, sequence and predicted structure of BRV and ToRSV translation-enhancing elements are a true reflection of the diversity of nepovirus translation mechanisms. Modular evolution of translation-enhancing elements even within a virus family is well documented, as exemplified by the IRESs in the family *Potyviridae* [24] or the CITEs in the family *Tombusviridae* and genus *Luteovirus* [25]. As shown for melon necrotic spot virus, inter-family exchange of CITEs can facilitate virus adaptation to new hosts, and even the breaking of plant resistance [77]. Thus, nepovirus RNAs may just be one more example of the modular evolution of translation mechanisms and the diversity of translation-enhancing elements.

## Supporting information

**S1 Table. Sequences and oligonucleotides used in the study.**
(DOCX)

**S2 Table. Putative translation initiation sites in selected nepovirus RNAs.** The nucleotide sequence surrounding in-frame AUG codons is shown (positions -3 to +5). Positions that correspond to the plant consensus start codon context: (G/A)(A/C)aAUGGC are underlined. C or U at the -3 position and/or A, C or U at the +4 position are rare in this consensus context and are highlighted in yellow. Predicted stem-loops downstream from AUGs are listed if they are less than 23 nts away from the AUG and have a free energy **ΔG** equal to or lower than -10. The distance to the AUG (in nts) and the calculated free energy (ΔG) of the stem-loops are indicated in the parentheses. Only the first AUG is listed if it is placed in an optimal conserved context or if a stem-loop with acceptable free energy is predicted at a location 10 to 23 nts downstream of the AUG. When the first AUG does not meet these conditions, subsequent AUGs within the first 1200 nts of the RNA are listed (up to the first four), until they meet these conditions. The NCBI accession number is shown for each sequence. Abbreviations are as follows: ToRSV (tomato ringspot virus), BRV (blackcurrant reversion virus), CLRV (cherry leaf roll virus), PRMV (peach rosette mosaic virus), BLSV (blueberry latent spherical virus), SLSV (soybean latent spherical virus), GBLV (grapevine Bulgarian latent virus), ArMV (arabis mosaic virus), GFLV (grapevine fanleaf virus), RRSV (raspberry ringspot virus), TRSV (tobacco ringspot virus), MMLRaV (mulberry mosaic roll leaf-associated virus), GDefV (grapevine deformation virus), PBRSV (potato black ringspot virus), MMMoV (melon mild mottle virus), TBRV (tomato black ring virus), CNSV (cycas necrotic stunt virus), AILV (artichoke Italian latent virus).
(DOCX)

**S1 Fig. Sequence alignment of the 5' region of the genomic RNAs of selected ToRSV isolates.** The sequences of ToRSV RNAs were aligned using Clustal Omega and alignment results are depicted for the ~900 initial nucleotides. The three putative start codons are shown in red. Sequences at the 5' end of the RNA complementary to predicted loops in region 3a-b are highlighted in light blue (see **Figs 9** and **S4** for more details). The predicted complementary stem sequences of a putative stem-loop (5' SL) structure located after the first AUG are shown with arrows. Labelling of ToRSV RNAs and NCBI accession numbers are as in **Fig 2**.
(DOCX)

**S2 Fig. Sequence alignment of the 3' UTRs of the genomic RNAs of selected ToRSV isolates.** The sequences of ToRSV RNAs 3' UTRs were aligned using Clustal Omega. Stop codons are indicated in red. The start point of each deletion mutant is indicated with small black arrows: R1 (region 1), R2 (region 2), R3/R3a (region 3 and region 3a), R3b (region 3b), R3c (region 3c), R3d (region 3d) and R4 (region 4). Sequences complementary to the 5' end of the RNA are highlighted in light blue (see Figs 9 and S4 for more details).
(DOCX)

**S3 Fig.** *In vitro* **translation rate of the VRV transcript at various transcript concentrations.** VRV transcripts were added to *in vitro* translation reactions at the concentrations of 0.05, 0.1, 0.2, 0.4, 0.8, 1, 2, 4 or 5 picomoles per 50 μl WGE and the resulting luminescence were measured. Experiments were repeated twice, each with three technical repeats, with similar results. A representative result is shown. Error bars represent the standard deviation of the three technical repeats.
(PPTX)

**S4 Fig. Predicted secondary structures of the 5' region and of region 3a-b of the 3' UTR for the RNAs of three representative ToRSV isolates.** The predicted secondary structures are shown for the RNAs of each isolate (Rasp1, 13C280, GYV) as indicated. The 5' region spans from the first nucleotide of the RNA to the nucleotide corresponding to the position of the third AUG codon in ToRSV-Rasp1 RNA2 (see alignment in **Fig 2**). Region 3a-b correspond to the region aligning with the corresponding region of ToRSV-Rasp1 RNA2 (see alignment in **S2 Fig**). Please note that regions 3a-b of ToRSV-GYV RNA1 and RNA2 are identical. In frame AUG codons are shown in green. The position of the 5' SL located downstream of the first AUG is shown for each RNA. Putative base pairing (sequence complementarity) between the 5' region and region 3a-b of the 3' UTR are indicated in light and dark blue. The light blue base pairing between a predicted loop in region 3a-b and the 5' end of the RNA is conserved for each isolate and is also shown in **Fig 9**. The putative dark blue base pairing between predicted exposed bulges in two stems is only conserved in isolates Rasp1 and 13C280 and is not shown in **Fig 9**. Secondary structures were predicted and visualized as described in **Fig 9**. Polypyrimidine stretches present in exposed loops or bulges in the predicted structures are highlighted in yellow.
(PPTX)

**S5 Fig. Phylogenetic analysis of the 3' UTR of nepovirus of subgroup C.** The nucleotide sequence of the 3' UTRs of RNA1 and RNA2 were aligned by Clustal W as implemented in MEGA X. Phylogenetic trees were generated using the maximum likelihood method and the validity of the branches was verified using 1000 bootstraps. For comparison, the deduced amino acid sequence of the Pro-Pol region (encoded by RNA1 and defined as the region between the catalytic cysteine of the protease and the GDD motif of the polymerase) and of the coat protein (encoded by RNA2) were also aligned by Clustal W and phylogenies were produced and tested as above.
(PPTX)

**S6 Fig. Predicted secondary structures of the 5' region and of selected regions of the 3' UTR of BRV RNA2.** (A) The 5' region spans from the first 440 nucleotides of the RNA corresponding to the 5' region shown for ToRSV RNAs in **Figs 9** and **S4**. Because the entire predicted secondary structure did not fit in the frame, nts 327–440 are shown above nts 1–327 but would be contiguous in the overall predicted secondary structure. In frame AUG codons are shown in green. The position of a previously identified 5' SL is shown [42]. The loop of this stem-loop (highlighted in light blue) is involved in an experimentally validated kissing loop

interaction with the loop of 3' SL-1 of the 3' UTR region A2 shown in (B). Another putative stem-loop located downstream of the first AUG (5' SL-2) and corresponding to the 5' SL of ToRSV RNAs is shown. Please note that the 5' coding region has not been studied for BRV and that the biological relevance of 5' SL-2 is not known. (B) Regions of the BRV RNA2 3' UTR previously identified as playing a role in translation of RNA2 [42], see **Fig 10A** for schematic representation of the position of these regions. The A2 region was shown to be the most critical. It contains two previously identified stem-loops, the first of which (3' SL-1) is involved in a kissing-loop interaction with the 5' SL (shown in light blue). Region C2 was shown to play a modest role in translation and includes a proposed alternative base pairing with the 5' SL (shown in light blue). Secondary structures were predicted and visualized as described in **Fig 9**. Polypyrimidine stretches present in exposed loops or bulges in the predicted structures are highlighted in yellow.
(PPTX)

**S1 Raw images.**
(PDF)

## Acknowledgments

We thank Drs. Eric Jan (University of British Columbia, Canada) and K. Andrew White (York University, Canada) for helpful discussions and critical review of the manuscript prior to submission. We are also grateful to the two reviewers for their constructive suggestions during the peer-review process.

## Author Contributions

**Conceptualization:** Dinesh Babu Paudel, Hélène Sanfaçon.

**Data curation:** Dinesh Babu Paudel, Hélène Sanfaçon.

**Formal analysis:** Dinesh Babu Paudel, Hélène Sanfaçon.

**Funding acquisition:** Hélène Sanfaçon.

**Investigation:** Dinesh Babu Paudel.

**Project administration:** Hélène Sanfaçon.

**Supervision:** Hélène Sanfaçon.

**Validation:** Dinesh Babu Paudel.

**Writing – original draft:** Dinesh Babu Paudel.

**Writing – review & editing:** Dinesh Babu Paudel, Hélène Sanfaçon.

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
