## [Decision Letter · Decision Letter 0]

18 Feb 2021

PONE-D-21-01859

Mapping of sequences in the 5’ region and 3’ UTR of tomato ringspot virus RNA2 that facilitate cap-independent translation of reporter transcripts in vitro

PLOS ONE

Dear Helene,

Thank you for submitting your manuscript to PLOS ONE. I now received both reviews on your manuscript and the comments are appended. As you can see, both reviewers recommended the manuscript as suitable for publication in PLOS ONE after minor revisions. Therefore, I invite you to submit a revised version of the manuscript that addresses the points raised during the review process.

We look forward to receiving your revised manuscript.

Kind regards,

A. L. N. Rao

Academic Editor

PLOS ONE

Reviewers' comments:

Reviewer's Responses to Questions

**Comments to the Author**

1. Is the manuscript technically sound, and do the data support the conclusions?

Reviewer #1: Yes

Reviewer #2: Yes

2. Has the statistical analysis been performed appropriately and rigorously? 

Reviewer #1: Yes

Reviewer #2: Yes

3. Have the authors made all data underlying the findings in their manuscript fully available?

Reviewer #1: Yes

Reviewer #2: Yes

4. Is the manuscript presented in an intelligible fashion and written in standard English?

Reviewer #1: Yes

Reviewer #2: Yes

5. Review Comments to the Author

Reviewer #1: This manuscript provides information on translation of nepoviruses, for which there is a scarcity of published research. The authors provide convincing evidence for a requirement of both the 5’ and the 3’ regions of ToRSV RNA 2 and have mapped a 3’ CITE to within a 200 nt region. While this information is interesting, it does not significantly advance what is already known for blackcurrant reversion virus. The paper could be much improved by providing more analysis of the 3’ CITE, which I believe could be obtained without additional experimentation for this report (important given the current research restrictions) and then further analyzed in a subsequent paper. There are also additional issues that will also need to be addressed before the paper is in an acceptable form.

Required improvements:

1. The authors provide putative structures of the 5’ and 3’ regions of RNA2. Good evidence is provided for one hairpin using co-variation analysis but not the rest of the structures. I believe that the authors need to extend this type of analysis (using covariation of the regions in other isolates and other related nepoviruses in both RNAs) to delineate the important conserved structures in the region. The authors already have familiarity with the downloaded version of RNA2drawer and they should switch to the online version (https://rna2drawer.app/) and use the pair complements mode to hand assemble the structure of these regions in other isolates/viruses. If the 3’CITE structure presented is accurate (to some extent), it will be present in the other isolates and most likely in the most closely related nepoviruses. The supplemental figures with mFold structures was not useful and should be omitted as the analysis was not conducted using the full-length virus and is not backed up by any other phylogenetic information. Since I couldn’t read the sequences in my printed version or on my high resolution computer screen, I had to redraw the structures to examine them, and I chose to use sequences from RNA1. Most of the structures shown in Figure 9 are also in ToRSV RNA1 (see attached RNA2drawer files- must be opened using the online app).

This analysis may also assist the authors in a better “guess” as to possible long-distance interaction sequences. Both of their proposed interactions have little chance of being correct due to their placement and sequence (there is almost never a G:U pair in these interactions and certainly never two of them). A much more likely interaction is between the terminal loop of their 5’ side hairpin (5’UUUCGCAA) and the very 5’ end of RNA 2 (3’ AAAGCGAA 5’ end) (the 5’ side hairpin atop a three-way junction structure is frequently engaged in such an interaction- see tombusvirus 3’ CITEs and PTE 3’ CITEs). However, if this latter possibility is not real, then deleting the sequence from the first AUG onward in construct V76RV would expose this 5’ end sequence, which will then pair with the 3’CITE and interfere with its function, depressing translation as shown in Figure 4. The authors should at least comment on this possibility.

2. It was nearly impossible to read the sequence figures (Figures 2 and 9). These need substantial improvement unless the issue was with the pdf conversion. Figure 9 should look much better if the export function (SVG or PPT) in RNA2Drawer was used (please see attached figures).

Minor issues:

1. line 84: region in the 5’ UTR

2. line 257: EtBr-stained

3. line 273: Why was it “interesting” that mutating AUG1 would result in a new band the size of initiation at AUG2? This is pretty much expected. What IS interesting is that the results are different for AGG1,2 and AGG 1,2,3. The result for AGG1,2 seems to indicate a low level of non-canonical initiation at AGG1 (also found for mutant AGG1). Why is this band absent in AGG1,2,3? This deserves a comment even if it is to say that it isn’t known why.

4. line 297 etc: Please correct throughout the following: it isn’t “the” translation of uncapped ARA transcript (or whatever). Its just: translation (no “the”) of the uncapped ARA transcript

5. Please include some numbering of deletion positions on Figure 8A.

Reviewer #2: This manuscript is technically sound, and the information is new, the data are clearly presented, so it is acceptable for publication in PLoS ONE. This work uses a high-fidelity translation system (wheat germ extract), to resolve some ambiguities based on sequence alone, of the start codon used by the Nepovirus, Tomato ringspot virus (ToRSV), and to identify sequences in the UTRs required for cap-independent translation.

The stimulation of translation of uncapped RNA containing viral UTRs, relative to mRNAs with nonviral UTRs is only 2 to 3-fold, so the effect is rather modest (Fig. 4B). Also, a concern is that the capped nonviral positive control gives less than 3-fold stimulation (with large error bars) over the uncapped nonviral construct (Fig. 4B), suggesting the capping reaction may have been inefficient. The reason(s) for this lack of stimulation by capping should be discussed.

Also, this work addresses the same question, RNA sequences that control cap-independent translation by nepoviruses, as previous papers by Karetnikov et al (refs 40-42) for different nepoviruses. This is mentioned in the Introduction, but I recommend a more detailed comparison of the results obtained here with the RNA structures of Karetnikov's work. As an interested reader, I found myself going to look up Karetnikov's results and trying to compare. In the Discussion, it would really flesh out this paper to compare structures and to explain how results were similar or different from those of Karetnikov.

Finally, given the low level of stimulation of translation by the UTRs, perhaps discuss the possible roles of the coding regions, and/or the VPg in directing translation initiation. Does the VPg interact with translation factors, like potyvirus and solemovirus VPgs do?

Specific details:

Line 72: Authors should note that, while most Potyvirid IRESes are relatively unstructured mostly (and all known IRESes in genus Potyvirus are unstructured), there is the notable exception of Triticum mosaic virus (genus Tritimovirus) which has a >700 nt, highly structured IRES resembling those of picornaviruses (see papers by Rakotondrafara lab).

Line 95: I would not consider a 76 nt 5' UTR to be short. Lots of viruses have shorter 5' UTRs. Saying shorter would be more appropriate.

Line 246: ...additional sequence alignments... (remove s).

Line 289: ...in more detail, we first... (remove s)

Line 530: ...Putative base pairing (sequence complementarity)...

Line 531: ...are indicated by bracket.

Line 611: ...Sequences on the predicted stem...

6. PLOS authors have the option to publish the peer review history of their article (what does this mean?). If published, this will include your full peer review and any attached files.

Reviewer #1: **Yes: **Anne E. Simon

Reviewer #2: **Yes: **W. Allen Miller

---

## [Author Response · Author response to Decision Letter 0]

24 Mar 2021

Point-by-point response to the reviewers comments

Reviewer #1: This manuscript provides information on translation of nepoviruses, for which there is a scarcity of published research. The authors provide convincing evidence for a requirement of both the 5’ and the 3’ regions of ToRSV RNA 2 and have mapped a 3’ CITE to within a 200 nt region. While this information is interesting, it does not significantly advance what is already known for blackcurrant reversion virus. The paper could be much improved by providing more analysis of the 3’ CITE, which I believe could be obtained without additional experimentation for this report (important given the current research restrictions) and then further analyzed in a subsequent paper. There are also additional issues that will also need to be addressed before the paper is in an acceptable form.

Response: We have added additional secondary structure analyses of the 3’ CITE and expanded the discussion on comparison to the previous reports on blackcurrant reversion virus (BRV) translation mechanisms to better highlight the similarities and differences between the two viruses. Please note that although the general conclusions are indeed similar for the two viruses, the deduced structure of the CITE and the location of this CITE in the 3’ UTR are different. Using the in vitro translation system, we performed additional experiments (trans-inhibition of VRV transcripts with cap analog, trans-inhibition of cARA transcripts with the 3’ UTR) that provide additional insights on viral RNA translation mechanisms and that were not previously performed for BRV.

Required improvements:

1. The authors provide putative structures of the 5’ and 3’ regions of RNA2. Good evidence is provided for one hairpin using co-variation analysis but not the rest of the structures. I believe that the authors need to extend this type of analysis (using covariation of the regions in other isolates and other related nepoviruses in both RNAs) to delineate the important conserved structures in the region. The authors already have familiarity with the downloaded version of RNA2drawer and they should switch to the online version (https://rna2drawer.app/) and use the pair complements mode to hand assemble the structure of these regions in other isolates/viruses. If the 3’CITE structure presented is accurate (to some extent), it will be present in the other isolates and most likely in the most closely related nepoviruses. The supplemental figures with mFold structures was not useful and should be omitted as the analysis was not conducted using the full-length virus and is not backed up by any other phylogenetic information. Since I couldn’t read the sequences in my printed version or on my high resolution computer screen, I had to redraw the structures to examine them, and I chose to use sequences from RNA1. Most of the structures shown in Figure 9 are also in ToRSV RNA1 (see attached RNA2drawer files- must be opened using the online app).

This analysis may also assist the authors in a better “guess” as to possible long-distance interaction sequences. Both of their proposed interactions have little chance of being correct due to their placement and sequence (there is almost never a G:U pair in these interactions and certainly never two of them). A much more likely interaction is between the terminal loop of their 5’ side hairpin (5’UUUCGCAA) and the very 5’ end of RNA 2 (3’ AAAGCGAA 5’ end) (the 5’ side hairpin atop a three-way junction structure is frequently engaged in such an interaction- see tombusvirus 3’ CITEs and PTE 3’ CITEs). However, if this latter possibility is not real, then deleting the sequence from the first AUG onward in construct V76RV would expose this 5’ end sequence, which will then pair with the 3’CITE and interfere with its function, depressing translation as shown in Figure 4. The authors should at least comment on this possibility.

Response: We thank the reviewer for her insightful comments. We have followed her suggestion and reanalyzed the secondary structure of ToRSV RNAs (RNA1 and RNA2 of three representative isolates, see new Fig 9 and S4 Fig). Please note that the isolates we used in this analysis are different from the reference isolate that the reviewer had used for the secondary structure predictions she kindly submitted along with her review. Thus the secondary structure prediction vary slightly from her predictions. We prefer not to use the reference isolate in this analysis, because this isolate (which was sequenced 30 years ago in our research centre) has been lost and can no longer be studied in planta. Instead, we are using isolates, including ToRSV-Rasp1, which are readily available in our lab and which were resequenced in their entirety in 2015.

As recommended by the reviewer, we use the RNA2drawer online app to enhance the visualization of the predicted structures and to predict sequence complementarity. We thank the reviewer for her insights in the requirements for sequence complementarities involved in long-distance interactions. We also thank her for identifying the sequence complementarity between the 5’ end of the RNA and one of the loops of the branched structure in region 3a-b. In fact, the 5’ end of the RNA is strictly conserved among all ToRSV RNAs. In addition, putative stem-loops that included region of sequence complementarity to the 5’ end of the RNA were identified in exposed loops of region 3a-b for all ToRSV RNAs analyzed, although the branched structure was not conserved for all isolates. We have modified the text accordingly, also incorporating her suggestion on the potential biological relevance of the sequence complementarity.

We considered analyzing related nepoviruses to find similar features in their 3’ UTRs as suggested by the reviewer. Unfortunately, nepoviruses are only distantly related to each other. The 3’ UTRs vary in size and are quite diverse in their sequence. In phylogenetic analyses of nepovirus 3’ UTRs, ToRSV isolates group in a separate branch. We now include a phylogenetic analysis to illustrate the diversity of nepovirus 3’ UTRs (new S4 Fig) and have added some comments in the text (introduction and discussion) to explain this diversity. Another difficulty in analyzing the 3’ UTRs of other nepoviruses is that the location of translation-enhancing elements within the UTR may not be the same for all nepoviruses as exemplified when comparing ToRSV and BRV (see new Fig 10). When comparing ToRSV and BRV we also note that the sequence and implied secondary structures of the identified translation-enhancing elements are very different. Because of these limitations, we feel that attempting to predict the position, sequence and structure of translation-enhancing elements in other nepoviruses of subgroup C would be challenging and highly speculative. In fact, our study did not support previous predictions by Karetnikov and colleagues on the position and sequence of ToRSV translation-enhancing elements in the 3’ UTR.

2. It was nearly impossible to read the sequence figures (Figures 2 and 9). These need substantial improvement unless the issue was with the pdf conversion. Figure 9 should look much better if the export function (SVG or PPT) in RNA2Drawer was used (please see attached figures). 

Response: We have improved the figures to enhance their readability.

Minor issues:

1. line 84: region in the 5’ UTR . Response: This has been corrected

2. line 257: EtBr-stained Response: This has been corrected

3. line 273: Why was it “interesting” that mutating AUG1 would result in a new band the size of initiation at AUG2? This is pretty much expected. What IS interesting is that the results are different for AGG1,2 and AGG 1,2,3. The result for AGG1,2 seems to indicate a low level of non-canonical initiation at AGG1 (also found for mutant AGG1). Why is this band absent in AGG1,2,3? This deserves a comment even if it is to say that it isn’t known why.

Response: We have added a sentence to highlight the presence of low level of non-canonical translation initiation at or near AGG 1 for some mutant but not for the triple mutant. We do not have an explanation for this observation.

4. line 297 etc: Please correct throughout the following: it isn’t “the” translation of uncapped ARA transcript (or whatever). It’s just: translation (no “the”) of the uncapped ARA transcript

Response: This has been corrected throughout the manuscript.

5. Please include some numbering of deletion positions on Figure 8A. 

Response: Numbering of deletion positions have been added to Fig 7A and Fig 8A. Please note that we are now referring to the numbering in relation to the entire RNA2 of ToRSV-Rasp1.

Reviewer #2: This manuscript is technically sound, and the information is new, the data are clearly presented, so it is acceptable for publication in PLoS ONE. This work uses a high-fidelity translation system (wheat germ extract), to resolve some ambiguities based on sequence alone, of the start codon used by the Nepovirus, Tomato ringspot virus (ToRSV), and to identify sequences in the UTRs required for cap-independent translation.

The stimulation of translation of uncapped RNA containing viral UTRs, relative to mRNAs with nonviral UTRs is only 2 to 3-fold, so the effect is rather modest (Fig. 4B). Also, a concern is that the capped nonviral positive control gives less than 3-fold stimulation (with large error bars) over the uncapped nonviral construct (Fig. 4B), suggesting the capping reaction may have been inefficient. The reason(s) for this lack of stimulation by capping should be discussed.

Response: We agree with the reviewer that the efficiency of capping may have varied from one experiment to another explaining the large error bar. We have added a comment in the text to that effect.

Also, this work addresses the same question, RNA sequences that control cap-independent translation by nepoviruses, as previous papers by Karetnikov et al (refs 40-42) for different nepoviruses. This is mentioned in the Introduction, but I recommend a more detailed comparison of the results obtained here with the RNA structures of Karetnikov's work. As an interested reader, I found myself going to look up Karetnikov's results and trying to compare. In the Discussion, it would really flesh out this paper to compare structures and to explain how results were similar or different from those of Karetnikov.

Response: We have expanded the discussion to increase more comparison of our results with those of Karetnikov. We also provide new figures to enhance this discussion (see new Fig 10 and S6 Fig). 

Finally, given the low level of stimulation of translation by the UTRs, perhaps discuss the possible roles of the coding regions, and/or the VPg in directing translation initiation. Does the VPg interact with translation factors, like potyvirus and solemovirus VPgs do?

Response: We have added a paragraph in the discussion. We do not have much information on the role of nepovirus VPgs in translation. We had previously identified an in vitro interaction between the VPg-protease intermediate polyprotein of ToRSV and eIF(iso)4E although the role of this interaction in translation is not clear and the interaction was mediated by the protease domain of the intermediate polyprotein. Please also note that the nepovirus VPgs are much smaller than those of potyviruses and sobemoviruses. They are similar to the picornavirus VPgs which are not known to play a role in translation and which are actually removed from viral RNAs prior to translation. We agree with the reviewer that this would be an interesting area of further research.

Specific details:

Line 72: Authors should note that, while most Potyvirid IRESes are relatively unstructured mostly (and all known IRESes in genus Potyvirus are unstructured), there is the notable exception of Triticum mosaic virus (genus Tritimovirus) which has a >700 nt, highly structured IRES resembling those of picornaviruses (see papers by Rakotondrafara lab).

Response: We have added a comment to acknowledge the different type of IRES in TriMV.

Line 95: I would not consider a 76 nt 5' UTR to be short. Lots of viruses have shorter 5' UTRs. Saying shorter would be more appropriate.

Response: We agree with this comment. The 5’ UTR is short compared to those of picornaviruses but is not short compared to those of many plant viruses. We have modified the sentence as suggested.

Line 246: ...additional sequence alignments... (remove s). Response: This has been corrected.

Line 289: ...in more detail, we first... (remove s). Response: This has been corrected.

Line 530: ...Putative base pairing (sequence complementarity)... Response: This has been corrected.

Line 531: ...are indicated by bracket. Response: The figure was changed according to the recommendation of Reviewer 1 and the legend was adjusted to reflect this change.

Line 611: ...Sequences on the predicted stem... Response: This has been corrected.

---

## [Editor Report · Decision Letter 1]

29 Mar 2021

Mapping of sequences in the 5’ region and 3’ UTR of tomato ringspot virus RNA2 that facilitate cap-independent translation of reporter transcripts in vitro

PONE-D-21-01859R1

Dear Dr. Sanfacon,

We’re pleased to inform you that your manuscript has been judged scientifically suitable for publication and will be formally accepted for publication once it meets all outstanding technical requirements.

Kind regards,

A. L. N. Rao

Academic Editor

PLOS ONE
---

## [Editor Report · Acceptance letter]

31 Mar 2021

PONE-D-21-01859R1 

Mapping of sequences in the 5’ region and 3’ UTR of tomato ringspot virus RNA2 that facilitate cap-independent translation of reporter transcripts *in vitro*

Dear Dr. Sanfaçon:

I'm pleased to inform you that your manuscript has been deemed suitable for publication in PLOS ONE. Congratulations! Your manuscript is now with our production department. 

Kind regards, 

on behalf of

Dr. A. L. N. Rao 

Academic Editor

PLOS ONE